# ProFITi: Probabilistic Forecasting of Irregular Time Series via Conditional Flows

## Abstract

Probabilistic forecasting of irregularly sampled multivariate time series with missing values is an important problem in many fields, including astronomy, finance, and healthcare. Traditional methods for this task often rely on differential equations based models and make an assumption on the target distribution. In recent years, normalizing flow models have emerged as a promising approach for density estimation and uncertainty quantification, offering a flexible framework that can capture complex dependencies. In this work, we propose a novel model ProFITi for probabilistic forecasting of irregular time series with missing values using conditional normalizing flows. In this approach, the model learns a joint probability distribution over the future values of the time series conditioned on the past observations and query (future) time-channel information. As components of our model, we introduce a novel invertible triangular attention layer, and an invertible non-linear activation function on and onto the whole real line. We conduct extensive experiments on $4$ datasets, and demonstrate that the proposed model, ProFITi, provides significantly better forecasting likelihoods compared to the existing baseline models. Specifically, on average, ProFITi provides $4$ times higher likelihood over the previously best model.

## 1 Introduction

Irregularly sampled multivariate time series with missing values (IMTS) are common in various real-world scenarios such as astronomy, health, and finance. While accurate forecasting of these IMTS is important for informed decision-making, estimating the uncertainty associated with these forecasts becomes crucial to avoid overconfidence. Ordinary Differential Equations (ODE) based models are widely used for the task. Besides the low computational efficiency, the current ODE-based models (Schirmer et al., 2022; De Brouwer et al., 2019; Biloš et al., 2021) offer only marginal likelihood estimates. In practical applications, joint distributions are desired to capture dependencies and study forecasting scenarios or trajectories.

In this study, we propose two novel components that can be used in flow models: a sorted invertible triangular attention layer (`SITA`) parametrized by conditioning input, and an invertible activation function, `Shiesh`, that is on and onto $\mathbb{R}$. We further propose a novel conditional flow model called `ProFITi`, for **Pro**babilistic **F**orecasting of **I**rregularly sampled Multivariate **Ti**me series. ProFITi is a permutation invariant model and designed to learn conditional permutation invariant structured distributions. It consists of several invertible blocks build using SITA and Shiesh functions. Being a Flow-based model, ProFITi can learn any random conditional joint distribution, while existing models (De Brouwer et al., 2019; Biloš et al., 2021) learn only Gaussian (similar to GLM in Figure 1).

Through extensive experiments, we demonstrate that our ProFITi achieves state-of-the-art probabilistic forecasting results for IMTS. Our contributions are as follows:

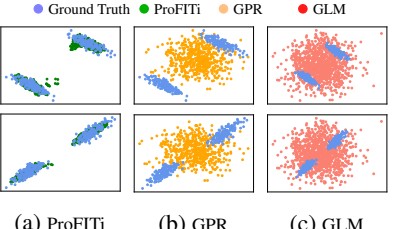

(a) ProFITi  (b) GPR  (c) GLM

Figure 1: Illustration of the predictions of our model ProFITi: to quantify uncertainty in a richer manner. Distribution generated by (a) ProFITi, (b) Gaussian Process Regression and (c) Generalized Linear Model. ProFITi provides a distribution close to the ground truth. See Section G.

Table 1: Summary of Important models that 1. can be applied to Time Series with irregular sampling (Irreg. Samp.), or missing values (Miss. Vals.), 2. can predict marginal distributions (Marg. Dist.) or joint distributions (Joint Dist.), 3. can learn on conditional densities (Condition), 4. density of sequences with variable lengths (Dynamic) or 5. Permutation Invariant (Perm. Inv.).

| Model | Irreg. Samp. | Miss. Vals. | Marg. Dist. | Joint Dist. | Condition | Dynamic | Perm. Inv. |
|---|---|---|---|---|---|---|---|
| GRU-ODE (De Brouwer et al., 2019) | ✓ | ✓ | (Param) | × | ✓ | ✓ | ✓ |
| Neural Flows (Biloš et al., 2021) | ✓ | ✓ | (Param) | × | ✓ | ✓ | ✓ |
| CRU (Schirmer et al., 2022) | ✓ | ✓ | (Param) | × | ✓ | ✓ | ✓ |
| GPR (Dürichen et al., 2015) | ✓ | ✓ | (Param) | (Param) | ✓ | ✓ | ✓ |
| GraFITi (Yalavarthi et al., 2023) | ✓ | ✓ | × | × | ✓ | ✓ | ✓ |
| RealNVP (Dinh et al., 2017) | × | × | × | ✓ | × | × | × |
| Inv. Autoreg (Kingma et al., 2016) | × | × | × | ✓ | × | × | × |
| Selv. Flow (van den Berg et al., 2018) | × | × | × | ✓ | × | × | × |
| Residual Flow (Behrmann et al., 2019) | × | × | × | ✓ | × | × | × |
| Graphical (Wehenkel & Louppe, 2021) | × | × | × | ✓ | × | × | × |
| Cond. NF Winkler et al. (2019) | × | × | × | ✓ | ✓ | × | × |
| Attn. Flow Sukthanker et al. (2022) | × | × | × | ✓ | ✓ | × | × |
| Inv. Dot. Attn Zha et al. (2021) | × | × | × | ✓ | ✓ | × | × |
| E(N) (Satorras et al., 2021) | × | × | × | ✓ | × | ✓ | ✓ |
| GNF (Liu et al., 2019) | × | × | × | ✓ | × | ✓ | ✓ |
| SNF (Biloš & Günnemann, 2021) | × | × | × | ✓ | × | ✓ | ✓ |
| MAF (Rasul et al., 2021) | × | × | ✓ | × | ✓ | ✓ | ✓ |
| CTFP (Deng et al., 2020) | ✓ | × | ✓ | × | ✓ | ✓ | ✓ |
| NKF (de Bézenac et al., 2020) | × | × | × | ✓ | ✓ | ✓ | ✓ |
| QFR (Si et al., 2022) | × | × | × | ✓ | ✓ | ✓ | ✓ |
| ProFITi (ours) | ✓ | ✓ | ✓ | ✓ | ✓ | ✓ | ✓ |

1. To the best of our knowledge, we are the first to investigate **normalizing flow based models for conditional permutation invariant structured distributions**. This makes them usable for probabilistic IMTS forecasting tasks.

2. We provide a novel invertible equivariant transformation, making the self attention mechanism invertible (in the last column), **sorted invertible triangular self attention**.

3. We provide a novel non-linear, invertible, differentiable activation function on and onto the whole real line, **Shiesh**. This activation function can be used in normalizing flows.

4. We provide a normalizing flow based model **ProFITi** for probabilistic forecasting of IMTS build from invertible self attention layers and transformation layers using Shiesh activation.

5. We conduct extensive experiments on 4 IMTS datasets, and show that our proposed model, ProFITi, significantly outperforms baselines in terms of normalized joint negative log-likelihood. On average, ProFITi provides 4 times higher likelihood over the previously best model. Implementation code: `anonymous.4open.science/r/ProFITi-8707/`.

## 2 LITERATURE REVIEW

**Probabilistic Forecasting Models for IMTS** Probabilistic IMTS forecasting often relies on variational inference or predicting distribution parameters. Neural ODE models (Chen et al., 2018) are popular, with VAE-based variants combining probabilistic latent states with deterministic networks. Other approaches like latent-ODE (Rubanova et al., 2019), GRU-ODE-Bayes (De Brouwer et al., 2019), Neural-Flows (Biloš et al., 2021), and Continuous Recurrent Units (CRUs; Schirmer et al. (2022)) have shown accurate results but provide only marginal distributions, no joint distributions. In contrast, Gaussian Process Regression (GPR) models (Dürichen et al., 2015; Li & Marlin, 2015; 2016; Bonilla et al., 2007) offer full joint posterior distributions for forecast outputs. However, multitask GPR can struggle due to positive definite covariance matrix constraints and computational demands on long time series data due to the matrix inverting operations. All the models assume data distribution is Gaussian and fail if the true distribution is different. However, ProFITi is a normalizing flow model, and is not constrained by this assumption (Kong & Chaudhuri, 2020).

**Normalizing Flows for variable input size** We deal with predicting distribution for variable many targets. This utilizes equivariant transformations, as shown in (Biloš & Günnemann, 2021; Satorras

et al., 2021; Liu et al., 2019). All the models apply continuous normalizing flows which require solving an ODE driven by a neural network. They tend to be slow due to the numerical integration process. Additionally, they cannot incorporate conditioning inputs.

**Conditioning Normalizing Flows**   The modeling of conditional data densities has been extensively explored within computer vision (Khorashadizadeh et al., 2023; Winkler et al., 2019; Anantha Padmanabha & Zabaras, 2021). They involve applying basic normalizing flow blocks such as affine transformations (Dinh et al., 2017), autoregressive transformations (Kingma et al., 2016) or Sylvester flow blocks (van den Berg et al., 2018). Often the conditioning values are appended to the target while passing through the flow layers as demonstrated in Winkler et al. (2019). For continuous data representations only a few works exist (Kumar et al., 2020; de Bézenac et al., 2020; Rasul et al., 2021; Si et al., 2022). However, methods that deal with regular multivariate time series (such as (Rasul et al., 2021)) cannot handle IMTS due to its variable size (in both prediction length and missing values). We solved this by combining attention (allowing flexible size) with invertibility (to be used in a normalizing flow).

**Flows with Invertible Attention**   To the best of our knowledge, there have been only two works that develop invertible attention for Normalizing Flows. Sukthanker et al. (2022) proposed an invertible attention mechanism by adding the identity matrix to a softmax attention. However, the major disadvantage of this mechanism is that softmax yields only positive values in the attention matrix and does not allow learning negative covariances. Zha et al. (2021) introduced residual attention similar to invertible residual flow (Chen et al., 2019; Behrmann et al., 2019). It has similar problems related to residual flows (Behrmann et al., 2019) such as the lack of an analytical inverse making inference slow as numerical methods have to be used. Additionally, these attention mechanisms often have problems with computing the determinant of the attention matrix as it has a complexity of $\mathcal{O}(K^3)$ with $K$ being the sequence length. We provide a summary of the related work in Table 1.

## 3   PROBLEM SETTING & ANALYSIS

**The IMTS Forecasting Problem.**   An **irregularly sampled multivariate times series with missing values** (called briefly just **IMTS** in the following), is a finite sequence $x^{\mathrm{obs}} = \left((t_i^{\mathrm{obs}}, c_i^{\mathrm{obs}}, o_i^{\mathrm{obs}})\right)_{i=1:I}$ of unique triples, where $t_i^{\mathrm{obs}} \in \mathbb{R}$ denotes the time, $c_i^{\mathrm{obs}} \in \{1, ..., C\}$ the channel and $o_i^{\mathrm{obs}} \in \mathbb{R}$ the value of an observation, $I \in \mathbb{N}$ the total number of observations across all channels and $C \in \mathbb{N}$ the number of channels. Let $\mathrm{TS}(C) := (\mathbb{R} \times \{1, \ldots, C\} \times \mathbb{R})^*$ denote the space of all IMTS with $C$ channels.[1]

An **IMTS query** is a finite sequence $x^{\mathrm{qu}} = \left((t_k^{\mathrm{qu}}, c_k^{\mathrm{qu}})\right)_{k=1:K} \in \mathrm{Q}(C) := (\mathbb{R} \times \{1, \ldots, C\})^*$ of just time points and channels (also unique), a sequence $y \in \mathbb{R}^K$ we call an answer and denote by $\mathrm{QA}(C) := \{(x^{\mathrm{qu}}, y) \in \mathrm{Q}(C) \times \mathbb{R}^* \mid |x^{\mathrm{qu}}| = |y|\}$ the space of all queries and possible answers.

The **IMTS probabilistic forecasting problem** then is, given a dataset $\mathcal{D}^{\mathrm{train}} := \left((x_n^{\mathrm{obs}}, x_n^{\mathrm{qu}}, y_n)\right)_{n=1:N} \in \mathrm{TS}(C) \times \mathrm{QA}(C)$ of triples of time series, queries and answers from an unknown distribution $p$ (with earliest query timepoint is beyond the latest observed timepoint for series $n$, $\min_k t_{n,k}^{\mathrm{qu}} > \max_i t_{n,i}^{\mathrm{obs}}$), to find a model $\hat{p}$ that maps each observation/query pair $(x^{\mathrm{obs}}, x^{\mathrm{qu}})$ to a joint density over answers, $\hat{p}(y_1, \ldots, y_K \mid x^{\mathrm{obs}}, x^{\mathrm{qu}})$, such that the expected negative log likelihood is minimal:

$$\ell^{\mathrm{NLL}}(\hat{p}; p) := -\mathbb{E}_{(x^{\mathrm{obs}}, x^{\mathrm{qu}}, y) \sim p} \log \hat{p}(y \mid x^{\mathrm{obs}}, x^{\mathrm{qu}})$$

Please note, that the number $C$ of channels is fixed, but the number $I$ of past observations and the number $K$ of future observations queried may vary over instances $(x^{\mathrm{obs}}, x^{\mathrm{qu}}, y)$. If query sizes $K$ vary, instead of (joint) negative log likelihood one also can normalize by query size to make numbers comparable over different series and limit the influence of large queries, the **normalized joint negative log likelihood** NJNL:

$$\ell^{\mathrm{NJNL}}(\hat{p}; p) := -\mathbb{E}_{(x^{\mathrm{obs}}, x^{\mathrm{qu}}, y) \sim p} \frac{1}{|y|} \log \hat{p}(y \mid x^{\mathrm{obs}}, x^{\mathrm{qu}}) \tag{1}$$

---

[1] $(\cdot)^*$ indicates the space of finite sequences of arbitrary length.

**Problem Analysis and Characteristics.** As the problem is not just an (unconditioned) density estimation problem, but the distribution of the outputs depends on both, the past observations and the queries, a **conditional density model** is required (**requirement 1**).

A crucial difference from many settings addressed in the related work is that we look for probabilistic models of the **joint distribution** of all queried observation values $y_1, \ldots, y_K$, not just at the **single variable marginal distributions** $p(y_k \mid x^{\mathrm{obs}}, x_k^{\mathrm{qu}})$ (for $k = 1{:}K$). The problem of marginal distributions is a special case of our formulation where all queries happen to have just one element (always $K = 1$).

So for joint probabilistic forecasting of IMTS, models need to output densities on a **variable number of variables** (**requirement 2**).

Furthermore, whenever two query elements get swapped, a generative model should swap its output accordingly, a density model should yield the same density values for outputs swapped the same way, i.e., the model should be **permutation invariant** (**requirement 3**): for every permutation $\pi$:

$$\hat{p}(y_1, \ldots, y_K \mid x^{\mathrm{obs}}, x_1^{\mathrm{qu}}, \ldots, x_K^{\mathrm{qu}}) = \hat{p}(y_{\pi(1)}, \ldots, y_{\pi(K)} \mid x^{\mathrm{obs}}, x_{\pi(1)}^{\mathrm{qu}}, \ldots, x_{\pi(K)}^{\mathrm{qu}}) \tag{2}$$

Note that permutation invariance is crucial w.r.t. channels. Permutation invariance w.r.t. time points is not a necessary property of a good model due to the causal nature of the series. But it is a very successful property of most state-of-the-art models in vision and time series (Zhou et al., 2021; Dosovitskiy et al., 2020; Chen et al., 2023). This motivated us to search for a novel invertible variant that can be used in conjunction with normalizing flows.

# 4 INVARIANT CONDITIONAL NORMALIZING FLOW MODELS

**Normalizing flows.** While parametrizing a specific distribution such as the Normal, is a simple and robust approach to probabilistic forecasting that can be added on top of any point forecasting model (for marginal distributions or fixed-size queries at least), such models are less suited for targets having a more complex distribution. Then typically normalizing flows are used (Rippel & Adams, 2013; Papamakarios et al., 2021). A normalizing flow is an (unconditional) density model for variables $y \in \mathbb{R}^K$ consisting of a simple base distribution, typically a standard normal $p_Z(z) := \mathcal{N}(z; 0_K, \mathbb{I}_{K \times K})$, and an invertible, differentiable, parametrized map $f(z; \theta)$; then

$$\hat{p}(y; \theta) := p_Z(f^{-1}(y; \theta)) \left| \det\left( \frac{\partial f^{-1}(y; \theta)}{\partial y} \right) \right| \tag{3}$$

is a proper density, i.e., integrates to 1, and can be fitted to data minimizing negative log likelihood via gradient descent algorithms. A normalizing flow can be conditioned on predictor variables $x \in \mathbb{R}^M$ by simply making $f$ dependent on predictors $x$, too: $f(z; x, \theta)$. $f$ then has to be invertible w.r.t. $z$ for any $x$ and $\theta$ (Trippe & Turner, 2018).

**Invariant conditional normalizing flows.** A conditional normalizing flow represents an invariant conditional distribution in the sense of eq. 2, if i) its predictors $x$ also can be grouped into $K$ elements $x_1, \ldots, x_K$ and possibly common elements $x^{\mathrm{com}}$: $x = (x_1, \ldots, x_K, x^{\mathrm{com}})$, and ii) its transformation $f$ is equivariant in stacked $x_{1:K}$ and $z$:

$$f(z^\pi; x_{1:K}^\pi, x^{\mathrm{com}}, \theta)^{\pi^{-1}} = f(z; x_{1:K}, x^{\mathrm{com}}, \theta) \quad \forall \text{permutations } \pi \tag{4}$$

where $z^\pi := (z_{\pi(1)}, \ldots, z_{\pi(K)})$ denotes a permuted vector. We call this an **invariant conditional normalizing flow model**. If $K$ is fixed, we call it **fixed size**, otherwise **dynamic size**. In IMTS forecasting, we have both inputs: $(x_1, \ldots x_K) := (x_1^{\mathrm{qu}}, \ldots, x_K^{\mathrm{qu}})$, $x^{\mathrm{com}} := x^{\mathrm{obs}}$.

**Invariant conditional normalizing flows via continuous flows.** Invariant conditional normalizing flow models have been developed in the literature based on the **continuous flow** approach (Chen et al., 2018; Grathwohl et al., 2019), where the transformation $f$ is specified implicitly by an ordinary differential equation with time-dependent vector field $g : \mathbb{R} \times \mathbb{R}^M \to \mathbb{R}^M$:

$$f^{-1}(z) := v(1) \quad \text{with } v : [0, 1] \to \mathbb{R}^K \text{ being the solution of } \frac{\partial v}{\partial \tau} = g(\tau, v(\tau)), \quad v(0) := z \tag{5}$$

$\tau$ often is called virtual time to clearly distinguish it from time as an input variable. The vector field $g$ is represented by a parametrized function $g(\tau, v; \theta)$ and then can be learnt. Continuous flow models can be made conditional by simply adding the predictors $x$ to the inputs of the vector field, too: $g(\tau, v; x, \theta)$. Unconditional structured continuous flow models can be made permutation invariant by simply making the vector field permutation equivariant (Köhler et al., 2020; Li et al., 2020; Biloš & Günnemann, 2021): $g(\tau, v^\pi; \theta)^{\pi^{-1}} = g(\tau, v; \theta)$. To make *conditional* structured continuous flow models permutation invariant, the vector field has to be **jointy permutation equivariant** in outputs $v$ and predictors $x$:

$$g(\tau, v^\pi; x^\pi, \theta)^{\pi^{-1}} = g(\tau, v; x, \theta)$$

The primary choice for a dynamic size, equivariant, parametrized function is self attention (SA):

$$A(X) := XW_Q(XW_K)^T, \quad A^{\text{softmax}}(X) := \text{softmax}(A(X))$$

$$\text{SA}(X) := A^{\text{softmax}}(X) \cdot XW_V$$

where $X$ is a matrix containing the elements $x_{1:K}$ as rows, $W_Q, W_K, W_V$ are parameter matrices (not depending on the number of rows of $X$) and the softmax is taken rowwise.

Self attention has been used in the literature as is for unconditional vector fields (Köhler et al., 2020; Li et al., 2020; Biloš & Günnemann, 2021). To be used in a conditional vector field, $X$ will have to contain both, the condition elements $x_{1:K}$ and the base samples $z_{1:K}$ stacked:

$$X := \begin{bmatrix} x_1^T & z_1 \\ \vdots & \vdots \\ x_K^T & z_K \end{bmatrix}$$

**Invariant conditional normalizing flows via invertible self attention.** When using self attention as vector field inside a continuous flow as in the previous section, then the continuous flow will provide invertibility. While an elegant and generic approach, continuous flows require ODE solvers and have been reported to be brittle and not straight-forward to train. We develop an alternative idea: to make self attention itself invertible (in the last column of $X$, which contains $z$). Then it can be used directly, without any need for an ODE wrapper. To get **invertible self attention (ISA)** (in the last column), we i) fix the last row of attention query and key matrices $W_Q$ and $W_K$ to zero, in effect *computing the attention matrix $A$ on the conditioners $x_{1:K}$ alone*, ii) fix all but the last rows of $W_V$ to zero and its last row to all ones, in effect *using the base sample $z_{1:K}$ alone as attention value*, and iii) regularize the attention matrix $A$ sufficiently to become invertible (see Section D):

$$A^{\text{reg}}(X) := \frac{1}{\|A(X)\|_2 + \epsilon} A(X) + \mathbb{I} \tag{6}$$

$$\text{ISA}(X) := A^{\text{reg}}(X_{:,1:|X|-1})X_{:,|X|} \tag{7}$$

where $\epsilon > 0$ is a hyperparameter. We note that different from a simple linear flow, the slope matrix $A^{\text{reg}}$ is not a parameter of the model, but computed from the conditioners $x_{1:K}$. Our approach is different from iTrans attention (Sukthanker et al., 2022, fig. 17) that makes attention invertible more easily via $A^{\text{iTrans}} := A^{\text{softmax}}(X) + \mathbb{I}$ using the fact that the spectral radius $\sigma(A^{\text{softmax}}(X)) \le 1$, but therefore is restricted to non-negative interaction weights.

The attention matrix $A^{\text{reg}}$ will be dense in general and thus slow to invert, taking $\mathcal{O}(K^3)$ operations. Following ideas for autoregressive flows and coupling layers, a triangular slope matrix would allow a much more efficient inverse pass, as its determinant can be computed in $\mathcal{O}(K)$ and linear systems can be solved in $\mathcal{O}(K^2)$. This does not restrict the expressivity of the model, as due to the Knothe–Rosenblatt rearrangement (Villani, 2009) from optimal transport theory, any two probability distributions on $\mathbb{R}^K$ can be transformed into each other by flows with a locally triangular Jacobian (where the local Jacobians in ODE models of optimal transport correspond to layers in the transformation of a normalized flow). Unfortunately, just masking the upper triangular part of the matrix will destroy the equivariance of the model. We resort to the simplest way to make a function equivariant: we sort the inputs before passing them into the layer and revert the outputs to the original ordering. We call this approach **sorted invertible triangular self attention (SITA)**:

$$\pi := \text{argsort}(x_1 S, \dots, x_K S) \tag{8}$$

$$A^{\text{tri}}(X) := \text{softplus-diag}(\text{lower-triangular}(A(X))) + \epsilon \mathbb{I} \tag{9}$$

$$\text{SITA}(X) := (A^{\text{tri}}(X_{:,1:|X|-1}^\pi)X_{:,|X|}^\pi)^{\pi^{-1}} \tag{10}$$

where $\pi$ operates on the rows of $X$. Sorting is a simple lexicographic sort along the dimensions of the $x_k S$. The matrix $S$ allows to specify a sorting criterion, e.g., a permutation matrix. For example, we will sort time series queries first by time stamp, so that the triangular matrix corresponds to a causal attention, second by channel, fixing some order of the channels.

## 5 A New Activation Function for Normalizing Flows

The transformation function $f$ of a normalizing flow usually is realized as a stack of several simple functions. As in any other neural network, elementwise applications of a function, called activation functions, is one of those layers that allows for non-linear transformations. However, most of the common activation functions used in deep learning such as ReLU are not applicable for normalizing flows, because they are not invertible (E1). Some like ELU cannot be used throughout the layer stack, because their output domain $\mathbb{R}^+$ does not cover all real numbers (E2). Some like Tanh-shrink ($\mathrm{tanhshrink}(u) \coloneqq u - \tanh(u)$) are invertible and cover the whole real line, but they have a zero gradient somewhere (for Tanh-shrink at 0) that will make computing the inverse of the normalizing factor $\left| \det\left( \frac{\partial f(u)}{\partial u} \right) \right|$ for the normalizing flow impossible (E3).

Table 2: Properties of existing activation functions.

To be used as a standalone layer in a normalizing flow, an activation function must fulfill these three requirements: **E1**. be invertible, **E2**. cover the whole real line and **E3**. have no zero gradients. Out of all activation functions in the pytorch library only Leaky-ReLU and P-ReLU meet all three requirements (see table 2). Both Leaky-ReLU and P-ReLU usually are used with a slope on their negative branch being well less than 1, so that stacking many of them might lead to small gradients also causing problems for the normalizing constant of a normalizing flow.

| Activation | E1 | E2 | E3 |
|---|---|---|---|
| ReLU | ✗ | ✗ | ✗ |
| Leaky-ReLU | ✓ | ✓ | ✓ |
| P-ReLU | ✓ | ✓ | ✓ |
| ELU | ✓ | ✗ | ✓ |
| SELU | ✓ | ✗ | ✓ |
| Swish | ✗ | ✗ | ✗ |
| GELU | ✗ | ✗ | ✓ |
| Tanh | ✓ | ✗ | ✓ |
| Sigmoid | ✓ | ✗ | ✓ |
| Tanh-shrink | ✓ | ✓ | ✗ |
| Shiesh | ✓ | ✓ | ✓ |

Unconstrained monotonic neural networks (Wehenkel & Louppe, 2019) have been proposed as versatile, learnable activation functions for normalizing flows, being basically a continuous normalizing flow for each scalar variable $u$ separately and a scalar field $g$ implemented by a neural network:

$$a(u) \coloneqq v(1) \quad \text{with } v : [0, 1] \to \mathbb{R} \text{ being the solution of } \frac{\partial v}{\partial \tau} = g(\tau, v(\tau)), \quad v(0) \coloneqq u \quad (11)$$

Lemma 2 (Section E) establishes the equivalence of unconstrained monotonic neural networks with continuous normalizing flows.

In consequence, they suffer from the same issues as any continuous normalizing flow: they are slow as they require explicit integration of the underlying ODE. Besides requirements E1–E3, activation functions will profit from further desired properties: **D1**. having an analytic inverse, **D2**. having an analytic Jacobian and **D3**. having a bounded gradient. Unconstrained Monotonic Neural Networks do not have desired property D1 and provide no guarantees for property D3.

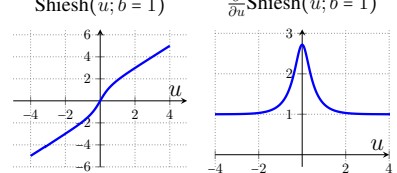

Figure 2: (left) Shiesh function, (right) partial derivative.

Instead of parametrizing the scalar field $g$ and learn it from data, we make an educated guess and choose a specific function with few parameters for which eq. 11 becomes explicitly solvable and requires no numerics at runtime: for the scalar field $g(\tau, a; b) \coloneqq \tanh(b \cdot a(\tau))$ the resulting ODE

$$\frac{\partial v}{\partial \tau} = \tanh(b \cdot v(\tau)), \quad v(0) \coloneqq u$$

has an explicit solution (Section F.1)

$$v(\tau; u, b) = \frac{1}{b} \sinh^{-1}\left( e^{b \cdot \tau} \cdot \sinh(b \cdot u) \right)$$

yielding our activation function Shiesh:

$$\mathrm{Shiesh}(u; b) \coloneqq a(u) \coloneqq v(1; u, b) = \frac{1}{b} \sinh^{-1}\left( e^b \cdot \sinh(b \cdot u) \right) \quad (12)$$

being invertible, covering the whole real line and having no zero gradients (E1–E3) and additionally with analytical inverse and gradient (D1 and D2)

$$\text{Shiesh}^{-1}(u;b) = \frac{1}{b}\sinh^{-1}\left(e^{-b}\cdot\sinh(b\cdot u)\right) \tag{13}$$

$$\frac{\partial}{\partial u}\text{Shiesh}(u;b) = \frac{e^{b}\cosh(b\cdot u)}{\sqrt{1+\left(e^{b}\sinh(b\cdot u)\right)^{2}}}$$

and bounded gradient (D3) in the range $(1, e^{b}]$ (Section F.4). Fig. 2 shows a function plot. In our experiments we fixed its parameter $b = 1$.

## 6 OVERALL PROFITI MODEL ARCHITECTURE

Invertible attention and the Shiesh activation function systematically model inter-dependencies between variables and non-linearity respectively, but do not move the zero point. To accomplish the latter, we use a third layer called **elementwise linear transformation layer** (EL):

$$\text{EL}(y_k; x_k) \coloneqq y_k\cdot\text{NN}^{\text{sca}}(x_k) + \text{NN}^{\text{trs}}(x_k) \tag{14}$$

where $\text{NN}^{\text{sca}}$ and $\text{NN}^{\text{trs}}$ are neural networks for scaling and translation. $\text{NN}^{\text{sca}}$ is equipped with a $\exp(\tanh)$ output function to make it positive and bounded, guaranteeing the inverse. We combine all three layers from eq. 7, 12, and 14 to a block

$$\text{profiti-block}(y; x) \coloneqq \text{Shiesh}(\text{EL}(\text{SITA}(z; x); x)) \tag{15}$$

and stack $L$ of those blocks to build the inverse transformation $f^{-1}$ of our conditional invertible flow model ProFITi. We add a transformation layer with slope fixed to 1 as initial encoding on the $y$-side of the model. See Figure 3 for an overview of its architecture.

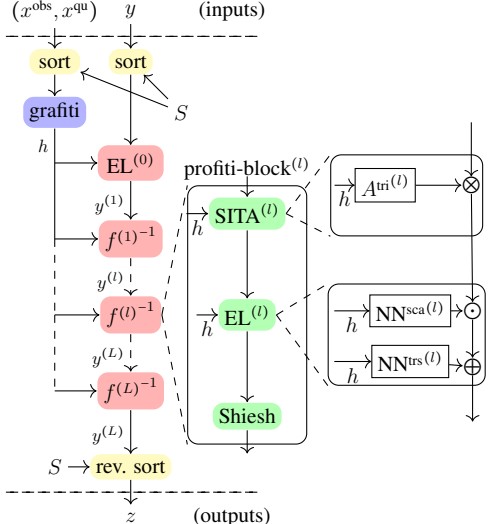

Figure 3: ProFITi architecture. $\otimes$: dot product, $\odot$: Hadamard product, and $\oplus$: addition. We reference the functions used to their equation numbers. $S$ (8), grafiti (16), SITA ( 10), EL (14), Shiesh (12)

**Query embedding.** As discussed in Section 4, for probabilistic time series forecasting we have to condition on both, the past observations $x^{\text{obs}}$ and the time point/channel pairs $x^{\text{qu}}$ of interest that are queried. While in principle any equivariant encoder could be used, an encoder that leverages the relationships between those two pieces of the conditioner is crucial. We use GraFITi (Yalavarthi et al., 2023), a graph based deterministic equivariant forecasting model for IMTS that provides state-of-the-art performance (in terms of accuracy and efficiency) as encoder

$$(h_1, \ldots, h_K) \coloneqq \text{GraFITi}(x_1^{\text{qu}}, \ldots, x_K^{\text{qu}}, x^{\text{obs}}) \tag{16}$$

The encoded conditioners $h_1, \ldots, h_K$ then are fed into ProFITi, i.e., take the roles of $x_1, \ldots, x_K$ in eq. 15. The Grafiti encoder is trained end-to-end within the Profiti model, we did not pretrain it.

Note that for each query, IMTS forecasting models yield a scalar, the predicted value, not an embedding vector. While it would be possible to use IMTS forecasting models as (scalar) encoders, due to their limitations to a single dimension we did not follow up on this idea.

**Training.** We train the ProFITi model $\hat{p}$ for the normalized joint negative log-likelihood loss (NJNL; eq. 1) which written in terms of the transformation $f^{-1}(\cdot; \cdot; \theta)$ of the normalizing flow and its parameters $\theta$ yields:

$$\ell^{\text{NJNL}}(\theta) \coloneqq \ell^{\text{NJNL}}(\hat{p}; p) \tag{17}$$

$$= -\mathop{\mathbb{E}}_{(x^{\text{obs}}, x^{\text{qu}}, y)\sim p}\frac{1}{|y|}\left(\sum_{k=1}^{|y|} p_Z(f^{-1}(y; x^{\text{obs}}, x^{\text{qu}}; \theta)_k) - \log\left|\det\left(\frac{\partial f^{-1}(y; x^{\text{obs}}, x^{\text{qu}}; \theta)}{\partial y}\right)\right|\right)$$

Table 3: Normalized Joint Negative Log-likelihood (NJNL), lower the better. Best is presented in bold. OOM indicates out of memory error. LL ratio ↑ shows the ratio of Normalized Joint Likelihood $e^{-\text{NJNL}}$ (not log-level NJNL) w.r.t the next best model. $\frac{\text{time}}{\text{epoch}}$ denotes runtime per epoch.

| | USHCN | $\frac{\text{time}}{\text{epoch}}$ | Physioinet'12 | $\frac{\text{time}}{\text{epoch}}$ | MIMIC-III | $\frac{\text{time}}{\text{epoch}}$ | MIMIC-IV | $\frac{\text{time}}{\text{epoch}}$ |
|---|---|---|---|---|---|---|---|---|
| GPR | 1.194±0.007 | 2s | 1.367±0.074 | 35s | 3.146±0.359 | 71s | 2.789±0.057 | 227s |
| HETAVE | 0.146±0.012 | 1s | 0.561±0.012 | 8s | 0.794±0.032 | 8s | OOM | – |
| GRU-ODE | 0.494±0.116 | 100s | 0.501±0.001 | 155s | 0.837±0.012 | 511s | 0.823±0.318 | 1052s |
| Neural-Flows | 0.550±0.019 | 21s | 0.496±0.001 | 34s | 0.835±0.014 | 272s | 0.689±0.087 | 515s |
| CRU | 0.633±0.023 | 35s | 0.741±0.001 | 40s | 1.090±0.001 | 131s | OOM | – |
| CNF+ | 0.937±0.044 | 24s | 1.057±0.007 | 210s | 1.123±0.005 | 347s | 1.041±0.010 | 577s |
| GraFITi+ | 0.270±0.048 | 3s | 0.367±0.021 | 32s | 0.695±0.019 | 80s | 0.287±0.040 | 84s |
| ProFITi (ours) | **-1.998±0.158** | 6s | **-0.766±0.038** | 59s | **-0.240±0.068** | 97s | **-1.856±0.051** | 123s |
| LL ratio ↑ | 8.4 | | 3.1 | | 2.5 | | 8.5 | |

## 7 EXPERIMENTS

### 7.1 EXPERIMENT FOR NORMALIZED JOINT NEGATIVE LOG-LIKELIHOOD

**Datasets.** For evaluating ProFITi we use 3 publicly available real-world medical IMTS datasets: **MIMIC-III** (Johnson et al., 2016), **MIMIC-IV** (Johnson et al., 2021), and **Physionet'12** (Silva et al., 2012). Datasets contain ICU patient records collected over 48 hours. The preprocessing procedures outlined in Yalavarthi et al. (2023), Scholz et al. (2023), Biloš et al. (2021) and De Brouwer et al. (2019) were applied to MIMIC-III and MIMIC-IV, esp. observations in MIMIC-III and MIMIC-IV were rounded to intervals of 30 minutes and 1 min, respectively. Physionet'12 was preprocessed according to Yalavarthi et al. (2023), Che et al. (2018), Cao et al. (2018) and Tashiro et al. (2021) to obtain hourly observations. We also evaluated on publicly available synthetic climate dataset **USHCN** (Menne et al., 2015). It consists of climate data observed for 150 years from 1218 weather stations in USA.

**Baseline Models. ProFITi** is compared to 3 probabilistic IMTS forecasting models: **CRU** (Schirmer et al., 2022), **Neural-Flows** (Biloš et al., 2021), and **GRU-ODE-Bayes** (De Brouwer et al., 2019). We also extend the state-of-the-art deterministic forecasting model GraFITi (Yalavarthi et al., 2023) to the probabilistic setting by also outputting an elementwise variance for parametrizing a normal distribution, called **GraFITi+**. GraFIT+ helps to disentangle lifts originating from GraFITi (encoder) and those originating from ProFITi. As often interpolation models can be used seamlessly for forecasting, too, we include **HETVAE** (Shukla & Marlin, 2022), a state-of-the-art probabilistic interpolation model, for comparison. Furthermore, we include Multi-task Gaussian Process Regression (**GPR**) (Dürichen et al., 2015) as a baseline able to provide joint densities. As there is no previous continuous normalizing flow model that works for dynamic size and conditioning input, we adapt the model from Biloš & Günnemann (2021) by making it conditional following eq. 4, using vanilla attention as vector field $g$ in eq. 5 as well as L2 regularization for its weights, called **CNF+**.

**Protocol.** We split the dataset into Train, Validation and Test in ratio 70:10:20, respectively. We select the hyperparameters from 10 random hyperparameter configurations based on their validation performance. We run 5 iterations with random seeds on the train dataset with the chosen hyperparameters. Following Biloš et al. (2021), De Brouwer et al. (2019) and Yalavarthi et al. (2023), we use the first 36 hours as observation range and forecast the next 3 time steps for medical datasets and first 3 years as observation range and forecast the next 3 time steps. All models are implemented in PyTorch and run on GeForce RTX-3090 and 1080i GPUs. We compare the models for Normalized Joint Negative Log-likelihood (NJNL) loss (eq. 1). Except for GPR, CNF+ and ProFITi, we take the average of the marginal negative log-likelihoods of all the observations in a series to compute NJNL for that series. We would like to note that we do not use CRPS, CRPS-sum that are widely used in regular multivariate time series as they cannot cover joint distributions.

**Results.** Table 3 demonstrates the Normalized Joint Negative Log-likelihood (lower the better) and run time per epoch for all the datasets. Best results are presented in bold. ProFITi outperforms

Table 4: Results for auxiliary experiments, Marginal Negative Log-likelihood (MNL), Mean Squared Error (MSE), lower the better. Comparing ProFITi with published results (in brackets): [†] from De Brouwer et al. (2019), [‡] from Biloš et al. (2021), and [♯] from Yalavarthi et al. (2023).

| | USHCN | | MIMIC-III | | MIMIC-IV | |
|---|---|---|---|---|---|---|
| | MNL | MSE | MNL | MSE | MNL | MSE |
| NeuralODE-VAE | (1.460±0.100[†]) | (0.960±0.110[†]) | (1.350±0.010[†]) | (0.890±0.010†) | – | – |
| Sequential-VAE | (1.370±0.006[†]) | (0.830±0.070[†]) | (1.390±0.070[†]) | (0.920±0.090[†]) | – | – |
| GRU-D | (0.990±0.070[†]) | (0.530±0.060[†]) | (1.160±0.050[†]) | (0.790±0.060[†]) | – | – |
| GRU-ODE | 0.776±0.172 | 0.410±0.106 | 0.839±0.030 | 0.479±0.044 | 0.876±0.589 | 0.365±0.012 |
| | (0.840±0.110[†]) | (0.430±0.070[†]) | (0.830±0.040[†]) | (0.480±0.010[†]) | (0.748±0.045[‡]) | (0.379±0.005) |
| Neural-Flows | 0.775±0.180 | 0.424±0.110 | 0.866±0.097 | 0.479±0.045 | 0.796±0.053 | 0.374±0.017 |
| | – | (0.414±0.102[♯]) | (0.781±0.041[‡]) | (0.499±0.004[‡]) | (0.734±0.054[‡]) | (0.364±0.008[‡]) |
| GraFITi+ | 0.462±0.122 | **0.256±0.027** | 0.657±0.040 | **0.401±0.028** | 0.351±0.045 | **0.233±0.005** |
| | – | (0.272±0.047[♯]) | – | (0.396±0.030[♯]) | – | (0.225±0.001[♯]) |
| ProFITi (ours) | **-1.717±0.143** | 0.413±0.185 | **0.511±0.068** | 0.474±0.049 | **-0.762±0.119** | 0.251±0.001 |

all the prior approaches with significant margin on all the three datasets. The next best performing model is GraFITi+. We note that although GPR is predicting joint likelihoods, it performs poorly, likely because of having very few parameters. We cannot run CRU for MIMIC-IV in our GPU with 48GB memory due to out of memory errors. Surprisingly, despite being also a flow model like ProFITi, CNF+ did not perform as well in our tasks. The performance gains w.r.t. the next best model shown in table 3 are for normalized joint likelihoods ($e^{-\text{NJNL}}$, no log-level), as for NJNL they would be not meaningful due to having 0 in the NJNL scale.

## 7.2 Auxiliary Experiment for Marginals and Point forecast

Existing models in the related work De Brouwer et al. (2019) and Biloš et al. (2021) cannot predict joint distributions, hence their evaluation is restricted to Marginal Negative Log-likelihood (MNL):

$$\ell^{\text{MNL}}(\hat{p}; \mathcal{D}^{\text{test}}) := -\frac{1}{\sum_{(x^{\text{obs}}, x^{\text{qu}}, y) \in D^{\text{test}}} |y|} \sum_{(x^{\text{obs}}, x^{\text{qu}}, y) \in D^{\text{test}}} \sum_{k=1}^{|y|} \log\left(\hat{p}\left(y_k \mid x^{\text{obs}}, x_k^{\text{qu}}\right)\right) \qquad (18)$$

For additional comparison with published results of the baselines, we evaluate ProFITi for MNL and MSE as well. Table 4 compares ProFITi with **GRU-ODE**, **Neural-Flows**, and **GraFITi+**. Results also include **NeuralODE-VAE** (Chen et al., 2018), **Sequential-VAE** (Krishnan et al., 2015; 2017) and **GRU-D** (Che et al., 2018) from the published sources. For ProFITi, to yield marginal distributions for MNL directly, we modify $A^{\text{tri}}$ by zeroing off-diagonal elements after training. Whereas for MSE, we sample 100 predictions together with their density values and output the one with the highest density value.

We run the experiment with the same protocol mentioned in the baseline papers. We see that ProFITi outperforms baseline models again. The gains provided by ProFITi in MNL is less pronounced than in NJNL as the model is designed and trained to learn joint distributions. Whereas for MSE, GraFITi that is trained for Gaussian MNL performs better than ProFITi. This is expected because Gaussian MNL has an MSE component in it. However, for (IMTS) probabilistic forecasting models the primary metric of interest is (marginal or normalized joint) negative log likelihood, where ProFITi performs the best (see previous subsection). In Section H, we have additional experiments on ablation studies and scalability.

## Conclusions

In this work, we propose a novel model ProFITi for probabilistic forecasting of irregularly sampled multivariate time series with missing values using conditioning normalizing flows. ProFITi is a permutation invariant normalizing flow model for conditional permutation invariant structured distributions. We propose two novel model components, sorted invertible triangular self attention and Shiesh activation function in order to learn any random target distribution. Our experiments on 3 IMTS datasets demonstrate that ProFITi provides better likelihoods than a state-of-the-art IMTS forecasting baselines.

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

Table 5: Statistics of the datasets used our experiments. Sparsity means the percentage of missing observations in the time series. Time Sparsity means the percentage of time steps missing after discretizing the time series.

| Name | #Samples | #Chann. | Max. len. | Max. Obs. | Sparsity | Time Sparsity |
|------|----------|---------|-----------|-----------|----------|---------------|
| USHCN | 1100 | 5 | 290 | 320 | 77.9% | 84.3% |
| Physionet'12 | 12,000 | 37 | 48 | 520 | 85.7% | 4.4% |
| MIMIC-III | 21,000 | 96 | 96 | 710 | 94.2% | 72.9% |
| MIMIC-IV | 18,000 | 102 | 710 | 1340 | 97.8% | 94.9% |

## A    IMPORTANT NOTATIONS

Here, we explain some important notations used in the paper.

- $|.|$: outer length of a sequence.
- $X_{:,1:|X|-1}$: all columns of $X$ except the last one
- $X_{:,|X|}$: last column of $X$
- $\pi$: permutations
- $\pi^{-1}$: inverse of permutations $\pi$
- $x^{\pi}$: application of permutation $\pi$ to vector $x$
- $x^{\pi^{-1}}$: application of permutation $\pi^{-1}$ to vector $x$, $(x^{\pi})^{\pi^{-1}} = x$

## B    DATASET DETAILS

Three datasets are used for evaluating the proposed model. Basic statistics of the datasets is provided in Table 5.

**Physionet2012 Silva et al. (2012)**   encompasses the medical records of 12,000 patients who were hospitalized in the ICU. During the initial 48 hours of their admission, 37 vital signs were measured. We follow the protocol used in previous studies Che et al. (2018); Cao et al. (2018); Tashiro et al. (2021); Yalavarthi et al. (2023). After pre-processing, dataset consists of hourly observations making a total of up to 48 observations in each series.

**MIMIC-III Johnson et al. (2016)**   constitutes a medical dataset containing data from ICU patients admitted to Beth Israeli Hospital. 96 different variables from a cohort of 18,000 patients were observed over an approximately 48-hour period. Following the preprocessing procedures outlined in (Biloš et al., 2021; De Brouwer et al., 2019; Yalavarthi et al., 2023), we rounded the observations to 30-minute intervals.

**MIMIC-IV Johnson et al. (2021)**   is an extension of the MIMIC-III database, incorporating data from around 18,000 patients admitted to the ICU at a tertiary academic medical center in Boston. Here, 102 variables are monitored. We followed the preprocessing steps of (Biloš et al., 2021; Yalavarthi et al., 2023) and rounded the observations to 1 minute interval.

**USHCN Menne et al. (2015)**   is a climate dataset consists of the measurements of 5 variables (daily temperatures, precipitation and snow) observed over 150 years from 1218 meteorological stations in the USA. We followed the same pre-processing steps given in (De Brouwer et al., 2019; Yalavarthi et al., 2023) and se- lected a subset of 1114 stations and an observation window of 4 years (1996-2000).

## C    IMPLEMENTING CNF+

We detail the CNF+ model that is used for comparison in Section 7.1. First, to the best of our knowledge, there exists no continuous normalizing flow that can be applied directly to the current

problem setup of predicting conditional density of sequences with variable lengths. Hence, inspired from the CNF proposed by (Biloš & Günnemann, 2021), we implement CNF+ that can be applied for our case. First, we concatenate the conditioning inputs $x$ and answers $y$. Used canonical dot product attention as the vector filed $g$ and $[x \| y]$ as $v(0)$ in eq. 5. The output of the continuous flow $v(1)$ is considered $z$.

## D  INVERTIBILITY OF $A^{\text{reg}}$

We prove that $A^{\text{reg}}$ presented in Section 4 is invertible.

**Lemma 1.** *For any $K \times K$ matrix $A$ and $\epsilon > 0$, the matrix $\mathbb{I}_K + \frac{1}{\|A\|_2 + \epsilon} A$ is invertible. Here, $\|A\|_2 := \max\limits_{x \neq 0} \frac{\|Ax\|_2}{\|x\|}$ denotes the spectral norm.*

*Proof.* Assume it was not the case. Then there exists a non-zero vector $x$ such that $(\mathbb{I}_K + \frac{1}{\|A\|_2 + \epsilon} A)x = 0$. But then $(\|A\|_2 + \epsilon)x = -Ax$, and taking the norm on both sides and rearranging yields $\|A\|_2 \geq \frac{\|Ax\|_2}{\|x\|_2} = \|A\|_2 + \epsilon > \|A\|_2$, contradiction! Hence the lemma. $\square$

## E  UNCONSTRAINED MONOTONIC NEURAL NETWORKS ARE JUST CONTINUOUS NORMALIZING FLOWS

Any unconstrained monotonic neural network (Wehenkel & Louppe, 2019) can equivalently be written as a standard continuous normalizing flow. To make our deduction of the Shiesh activation function in section slightly more streamlined, we therefore have presented unconstrained monotonic neural network as continuous normalizing flows from the beginning.

**Lemma 2.** *Any UMNN function $a^{\text{UMNN}}$ defined by*

$$a^{\text{UMNN}}(u) := \int_0^u f(\tau) d\tau + b$$

*with a positive function $f$ can be represented as a continuous normalizing flow for a suitable scalar field $g$:*

$$a^{\text{CNF}}(u) := v(1) \quad \text{with } v : \mathbb{R} \to \mathbb{R} \text{ being the solution of } \frac{\partial v}{\partial \tau} = g(\tau, v(\tau)), \quad v(0) := u$$

*Proof.* Let $a^{\text{UMNN}} : \mathbb{R} \to \mathbb{R}$, $u \mapsto a^{\text{UMNN}}(u)$ be a UMNN function. The continuous normalizing flow to be constructed must connect each $(0, u)$ by a flowline to $(1, a^{\text{UMNN}}(u))$ in the product space $[0, 1] \times \mathbb{R}$ (see Figure 4). The easiest way to do this is via a line segment, namely the flowline

$$\phi_t(u) = (0, u) + t(1, a^{\text{UMNN}}(u) - u) , \ t \in [0, 1] .$$

These lines do not intersect, as for all $t \in [0, 1]$ the second coordinate is strictly monotounously increasing with respect to $u$: $\frac{\partial \phi_t(u)}{\partial u} = (0, (1 - t) + a^{\text{UMNN}'}(u)), (1 - t) + a^{\text{UMNN}'}(u) > 0$ (by UMNN).

We now have to write the curves $\phi_t(u)$ as integral curves of a time dependent vector field $g(t, u)$ on $[0, 1] \times \mathbb{R}$, or rather the second component of $\phi_t(u)$, namely $a_t(u) = u(1 - t) + a^{\text{UMNN}}(u)t$ as the solution of a differential equation

$$\frac{\partial a_t(u)}{\partial t} = g(t, a_t(u)) = g(\phi_t(u)) .$$

Now $\frac{\partial a_t(u)}{\partial t} = a^{\text{UMNN}}(u) - u$, so we have to set $g(\phi_t(u)) := a^{\text{UMNN}}(u) - u$ for any $t \in [0, 1]$ and $u \in \mathbb{R}$. We can explicitly write $g(t, u) = a^{\text{UMNN}}(a_t^{-1}(u)) - a_t^{-1}(u)$. The inverse of the continuously differentiable function $a_t$ with positive derivative exists and is continuously differentiable. $\square$

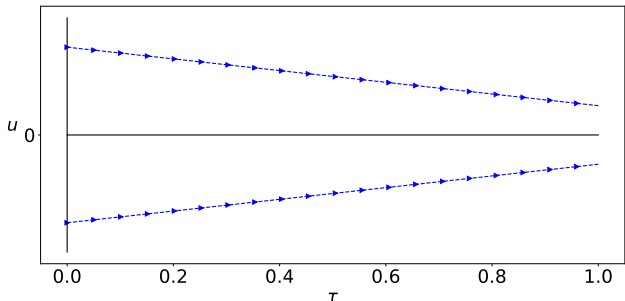

Figure 4: Demonstrating UMNN flowlines

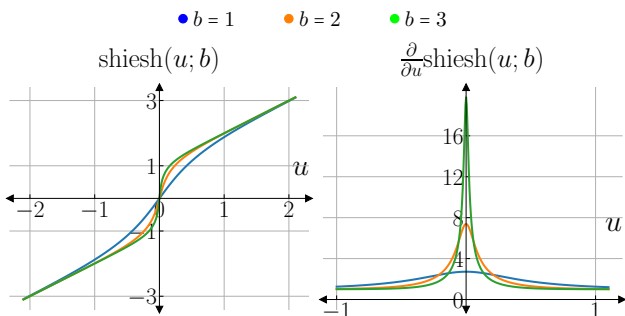

Figure 5: Demonstration of Shiesh activation function with varying $b$.

## F  Shiesh ACTIVATION FUNCTION

### F.1  SOLVING ODE

The differential equation $\frac{dv(\tau)}{d\tau} = \tanh(bv(\tau)), \quad v(0) := u$ can be solved by separation of variables. However, we can also proceed as follows by multiplying the equation with $b\cosh(b \cdot v(\tau))$:

$$b\cosh(bv(\tau))\frac{dv(\tau)}{d\tau} = b\sinh(bv(\tau))$$

$$\Leftrightarrow \qquad \frac{d\sinh(bv(\tau))}{d\tau} = b\sinh(bv(\tau))$$

$$\Leftrightarrow \qquad \sinh(bv(\tau)) = Ce^{b\tau} \quad \text{for some } C$$

$$\Leftrightarrow \qquad v(\tau) = \frac{1}{b}\sinh^{-1}(Ce^{b\tau}) \quad \text{for some } C .$$

The initial condition yields $C = \sinh(bu)$

### F.2  INVERTIBILITY OF Shiesh

A function $F : \mathbb{R} \to \mathbb{R}$ is invertible if it is strictly monotonically increasing.

**Theorem 1.** *Function Shiesh$(u;b) = \frac{1}{b}\sinh^{-1}(e^b\sinh(b \cdot u))$ is strictly monotonically increasing for $u \in \mathbb{R}$.*

*Proof.* A function is strictly monotonically increasing if its first derivate is always positive. From eq. 13, $\frac{\partial}{\partial u}\text{Shiesh}(u;b) := \frac{e^b\cosh(b \cdot u)}{\sqrt{1+(e^{b \cdot \tau}\sinh(b \cdot u))^2}}$. We known that $e^{b \cdot \tau}$ and $\cosh(u)$ are always positive hence $\frac{\partial}{\partial u}\text{Shiesh}(u;b)$ is always positive. $\qquad\square$

### F.3 IMPLEMENTATION DETAILS

Implementing Shiesh on the entire $\mathbb{R}$ will have numerical overflow. Hence, we implement it in piece-wise manner. In this work, we are interested in $b > 0$ and show all the derivations for it.

With $\sinh(x) = \frac{e^x - e^{-x}}{2}$ and $\sinh^{-1}(x) = \log(x + \sqrt{1 + x^2})$ Shiesh can be rewritten as follows:

$$
\begin{aligned}
\text{Shiesh}(u; b) &:= \frac{1}{b} \sinh^{-1}\left( \exp(b) \cdot \sinh(b \cdot u) \right) \\
&= \frac{1}{b} \log\left( \exp(b) \cdot \sinh(b \cdot u) + \sqrt{1 + \left( \exp(b) \cdot \sinh(b \cdot u) \right)^2} \right) \\
&= \frac{1}{b} \log\left( \left( \exp(b) \cdot \frac{\exp(b \cdot u) - \exp(-b \cdot u)}{2} \right) + \sqrt{1 + \left( \exp(b) \cdot \frac{\exp(b \cdot u) - \exp(-b \cdot u)}{2} \right)^2} \right)
\end{aligned}
$$

When $u \gg 0$, Shiesh can be approximated to the following:

$$
\begin{aligned}
\text{Shiesh}(u; b) &\approx \frac{1}{b} \log\left( \exp(b) \cdot \left( \frac{\exp(b \cdot u)}{2} \right) + \sqrt{1 + \left( \exp(b) \cdot \frac{\exp(b \cdot u)}{2} \right)^2} \right), \qquad \exp(-b \cdot u) \to 0 \\
&\approx \frac{1}{b} \log\left( \exp(b) \cdot \frac{\exp(b \cdot u)}{2} + \exp(b) \cdot \frac{\exp(b \cdot u)}{2} \right), \qquad \sqrt{1 + u^2} \approx u \quad \text{for} \quad u \gg 0 \\
&= \frac{1}{b} \log\left( \exp(b) \cdot \exp(b \cdot u) \right) \\
&= \frac{1}{b} \log\left( \exp(b) \right) + \frac{1}{b} \log\left( \exp(b \cdot u) \right) \\
&= 1 + u
\end{aligned}
$$

Now for $u \ll 0$, we know that $\sinh^{-1}(u)$ and $\sinh(u)$ are odd functions meaning

$$
\sinh^{-1}(-u) = -\sinh^{-1}(u) \tag{19}
$$
$$
\sinh(-u) = -\sinh(u) \tag{20}
$$

Also, we know that composition of two odd functions is an odd function making Shiesh an odd function. Now,

$$
\begin{aligned}
\text{Shiesh}(u; b) &\approx u + 1 && \text{for} && u >> 0 \\
\implies \text{Shiesh}(u; b) &\approx -(-u + 1) && \text{for} && u << 0
\end{aligned}
$$

Hence, to avoid numerical overflow in implementing Shiesh, we apply it in piece-wise manner as follows:

$$
\text{Shiesh}(u; b) = \begin{cases} \frac{1}{b} \sinh^{-1}(\exp(b) \sinh(b \cdot u)) & \text{if} \quad |x| \le 5 \\ u + 1 \cdot \text{sign}(u) & \text{else} \end{cases}
$$

Similarly, its partial derivative is implemented using:

$$
\frac{\partial}{\partial u} \text{Shiesh}(u; b) = \begin{cases} \frac{e^b \cosh(b \cdot u)}{\sqrt{1 + \left( e^b \sinh(b \cdot u) \right)^2}} & \text{if} \quad |x| \le 5 \\ 1 & \text{else} \end{cases} \tag{21}
$$

### F.4 BOUNDS OF THE DERIVATIVES

Assume $\textbf{DShiesh}(u; b) = \frac{\partial}{\partial u} \text{Shiesh}(u; b)$ and $b > 0$. For larger values of $u$, from eq. 21, $\textbf{DShiesh}(u; b) \approx 1$. Now we show that for the values $u \in [-5, 5]$ the maximum value for

```python
class shiesh(nn.Module):
    threshold: Tensor
    slope: Tensor

    def __init__(self) -> None:
        super().__init__()
        self.register_buffer("threshold", 5)
        self.register_buffer("slope", torch.exp(1.))

    def shiesh_(self, x: Tensor) -> Tensor:
        return torch.archsinh(torch.exp(1)*torch.sinh(x))

    def Dshiesh(self, x: Tensor) -> Tensor:
        return torch.exp(1)*torch.cosh(x)/(1 + (torch.exp(1)*torch.sinh(x))**2)**0.5

    def forward(self, x: Tensor) -> (Tensor, Tensor):
        mask = x.abs() <= self.threshold
        x_out =  torch.where(mask, self.shiesh_(x), x + torch.sign(x))
        ldj = torch.log(torch.where(mask, self.Dshiesh(x), x + torch.sign(x)))
        return x_out, ldj
```

(a) Pytorch implementation of Shiesh.

```python
class shiesh_inv(nn.Module):
    threshold: Tensor
    slope: Tensor

    def __init__(self) -> None:
        super().__init__()
        self.register_buffer("threshold", 5)
        self.register_buffer("slope", torch.exp(1.))

    def shiesh_inv_(self, x: Tensor) -> Tensor:
        return torch.archsinh(torch.exp(-1)*torch.sinh(x))

    def Dshiesh_inv(self, x: Tensor) -> Tensor:
        return torch.exp(-1)*torch.cosh(x)/(1 + (torch.exp(-1)*torch.sinh(x))**2)**0.5

    def forward(self, x: Tensor) -> (Tensor, Tensor):
        mask = x.abs() <= self.threshold
        x_out =  torch.where(mask, self.shiesh_inv_(x), x - torch.sign(x))
        ldj = torch.log(torch.where(mask, self.Dshiesh_inv(x), x + torch.sign(x)))
        return x_out, ldj
```

(b) Pytorch implementation of Shiesh$^{-1}$ .

Figure 6: Implementation of Shiesh and its inverse in Pytorch.

$\mathbf{D}\text{Shiesh}(u; b)$. For this we compute $\mathbf{D}^2\text{Shiesh}(u; b)$:

$$\mathbf{D}^2\text{Shiesh}(u; b) := \frac{be^b \sinh(bu) \left( e^{2b} \sinh^2(bu) - e^{2b} \cosh^2(bu) + 1 \right)}{\left( e^{2b} \sinh^2(bu) + 1 \right)^{3/2}}$$

$$:= \frac{be^b \sinh(bu)(1 - e^{2b})}{\left( e^{2b} \sinh^2(bu) + 1 \right)^{3/2}}$$

In order to compute the maximum of the function $\mathbf{D}\text{Shiesh}(u; b)$, we equate $\mathbf{D}^2\text{Shiesh}(u; b)$ to zero:

$$be^b \sinh(bu)(1 - e^{2b}) = 0 \quad \left( \left( e^{2b} \sinh^2(bu) + 1 \right)^{3/2} > 0 \right)$$

$$\implies \sinh(bu) = 0$$

$$\implies u = 0$$

Now, we compute $\mathbf{D}^3\text{Shiesh}(u; b)$ for $u = 0$. $\mathbf{D}^3\text{Shiesh}(u; b)$ can be given as:

$$\mathbf{D}^3\text{Shiesh}(u; b) = -\frac{b^2 e^b (2e^{2b} \sinh^2(bu) - 1) \cosh(bx)(e^{2b} \sinh^2(bu) - e^{2b} \cosh^2(bu) + 1)}{(e^{2b} \sinh^2(bu) + 1)^{5/2}}$$

Substituting $u = 0$, we get

$$\mathbf{D}^3\text{Shiesh}(0; b) = -\frac{b^2 e^b (2e^{2b} \cdot 0 - 1) \cdot 1 \cdot (e^{2b} \cdot 0 - e^{2b} \cdot 1 + 1)}{(e^{2b} \cdot 0 + 1)^{5/2}}$$

$$= b^2 e^b (1 - e^{2b}) \ < 0 \qquad (b > 0)$$

Hence, the bounds for the $\mathbf{D}\text{Shiesh}(u; b)$ is $\{1, e^b\}$.

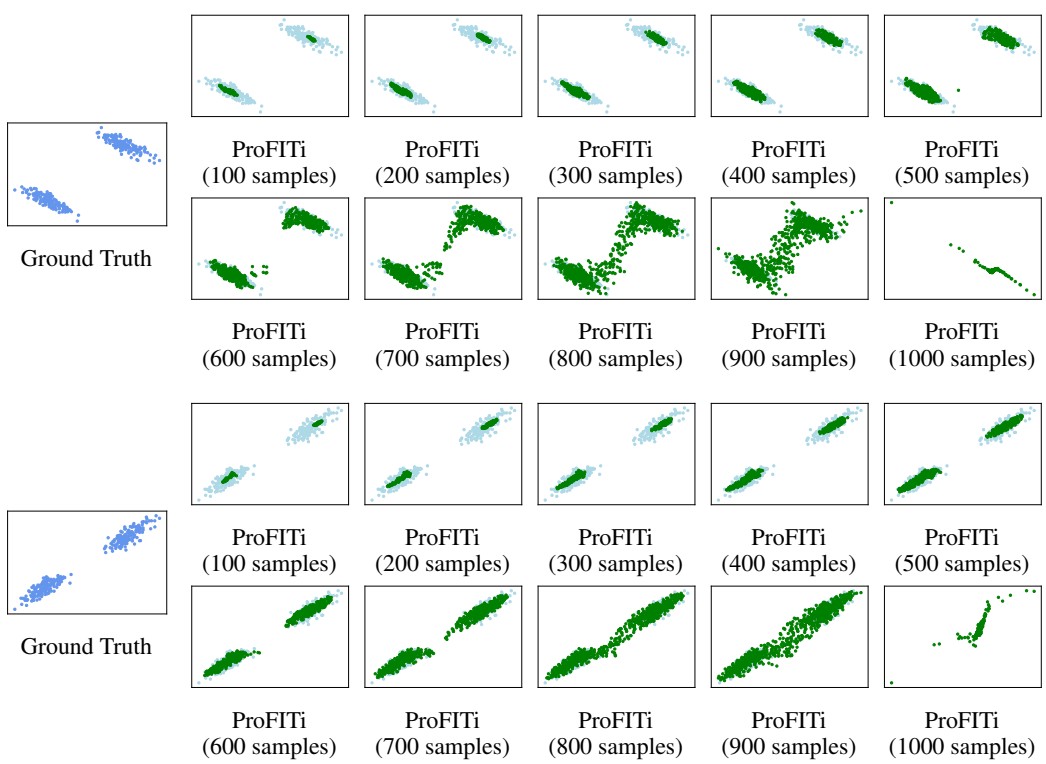

Figure 7: Demonstrating the distributions generated by ProFITi. MC sampling of 1000, sorted them with increasing likelihood. With increase in samples after sorting, the distribution deviates from the true distribution. For the images showing distributions of ProFITi, Ground Truth distribution is shown in the background.

## G CREATING TOY EXAMPLE FOR CONDITIONAL HETEROSCEDASTIC DISTRIBUTIONS: FIGURE 1

Here, we show how to generate the toy example used in the Section 1. It is a mixture of two bivariate Gaussian distributions. We first generate the conditioning variables $x_k \sim \mathbb{N}(0,1), k = 1:2$ (Eq. 22). Then, we use the generated $x$ to create a covariance matrix $\Sigma$ (Eq. 23). Now, we draw the samples using the mixture of Gaussians as in Eq 24. We allow large gap between two Gaussians so that the plots can look separable.

$$x_k \sim \mathcal{N}(0,1), \quad k \in 1{:}2, \quad x^{\text{com}} := () \tag{22}$$

$$\Sigma(x) := \begin{pmatrix} 1 + |x_1| & \operatorname{sgn}(x_1 x_2)\sqrt{(1+|x_1|)(1+|x_2|)-1} \\ \operatorname{sgn}(x_1 x_2)\sqrt{(1+|x_1|)(1+|x_2|)-1} & 1 + |x_2| \end{pmatrix} \tag{23}$$

$$y_{1:2} \sim \mathcal{N}\left(\begin{bmatrix} 5 + x_{1,1} \\ 5 + x_{2,1} \end{bmatrix}), \Sigma(x_{.,2})\right) + \mathcal{N}\left(\begin{bmatrix} -5 + x_{1,1} \\ -5 + x_{2,1} \end{bmatrix}), \Sigma(x_{.,2})\right) \tag{24}$$

For GPR, we implemented (Dürichen et al., 2015), whereas for Generalized Linear Model, we simply pass the $(x_1, x_2)$ to a single layer feed forward neural network and predicted mean and standard deviation of a normal distribution.

In Figure 7, we demonstrate the density generated by ProFITi. We randomly generated 1000 samples and sorted them according to their likelihoods. Then, we plot the density of those sorted samples in the increase order. As expected with all the samples (ProFITi (1000 samples)), samples with least likelihood will fall far outside the true distribution.

## H ADDITIONAL EXPERIMENTS

### H.1 ABLATION STUDIES: VARYING MODEL COMPONENTS

We show the impact of different ProFITi components using Physionet'12. We see that the Shiesh activation function provides a significant improvement as it can help learning non-Gaussian distributions (compare ProFITi and ProFITi-Shiesh). Similarly, learning joint distributions (ProFITi) provides better NJNL compared to ProFITi-SITA. Learning only Gaussian marginal distributions (ProFITi-SITA-Shiesh) performs significantly worse than ProFITi. Using PReLU instead of Shiesh (ProFITi-Shiesh+PReLU) deteriorates the performance of ProFITi. Using Leaky-ReLU leads to very small Jacobians and also has a vanishing gradient problem. We see that $A^{\text{iTrans}}$ (ProFITi-$A^{\text{tri}} + A^{\text{iTrans}}$) perform bad as it can learn only positive covariances. Finally, we see that ProFITi with either $A^{\text{reg}}$ or $A^{\text{tri}}$ per-

Table 6: Varying model components. Shown is NJNL. ProFITi-A+B indicates component A is removed and B is added.

| Model | Physionet2012 |
|---|---|
| ProFITi | -0.766±0.038 |
| ProFITi-SITA | -0.470±0.017 |
| ProFITi-Shiesh | 0.285±0.061 |
| ProFITi-SITA-Shiesh | 0.372±0.021 |
| ProFITi-Shiesh+PReLU | 0.384±0.060 |
| ProFITi-$A^{\text{tri}}$+$A^{\text{iTrans}}$ | -0.199±0.141 |
| ProFITi-$A^{\text{tri}}$+$A^{\text{reg}}$ | -0.778±0.016 |

forms comparably, however, $A^{\text{reg}}$ has scalability problems as computing the determinant of the full attention matrix has computational complexity $\mathcal{O}(K^3)$, while for the triangular attention matrix only $\mathcal{O}(K)$. Also, it performs worse with increasing forecast lengths (see Section H.3). We tried Leaky-ReLU instead of PReLU for the study but due to very small slope (0.01) for the negative values, it suffers from the vanishing gradient problem. Therefore no results are shown.

### H.2 EXPERIMENT ON VARYING THE ORDER OF THE CHANNELS

In ProFITi, we fix the order of channels to make SITA equivariant. In Figure 8, through critical difference diagram, we demonstrate that changing the permutation used to fix the channel order does not provide statistically significant difference in the results. ProFITi$-\pi_{1:5}$ indicate ProFITi with 5 different pre-fixed permutations on channels while time points are left in causal order. The order in which we sort channels and time points is a hyperparameter. To avoid this hyperparamerter and even allow different sorting criteria for different instances, one can parametrize $P_\pi$ as a function of $X_{1:|X|-1,.}$ (**learned**

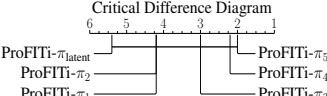

Figure 8: Statistical test on the results of various channel orders for ProFITi.

Table 7: Varying observation and forecast horizons of Physionet'12 dataset

| | obs/forc : 36/12hrs | | | obs/forc : 24/24hrs | | | obs/forc : 12/36hrs | | |
| | NJNL | run time (s) | | NJNL | run time (s) | | NJNL | run time (s) | |
| | | epoch | $A$ | | epoch | $A$ | | epoch | $A$ |
|---|---|---|---|---|---|---|---|---|---|
| Neural Flows | 0.709±0.483 | 109.6 | - | 1.097±0.044 | 46.6 | - | 1.436±0.187 | 45.5 | - |
| GraFITi+ | 0.522±0.015 | 42.9 | - | 0.594±0.009 | 43.1 | - | 0.723±0.004 | 37.5 | - |
| ProFITi | -0.768±0.041 | 64.8 | 3.3 | -0.355±0.243 | 66.2 | 5.2 | -0.291±0.415 | 82.1 | 8.6 |
| ProFITi-$A^{\text{tri}}$+$A^{\text{reg}}$ | -0.196±0.096 | 89.9 | 7.1 | 0.085±0.209 | 142.1 | 30.1 | 0.092±0.168 | 245.8 | 73.1 |

**sorted triangular invertible self attention**). ProFITi$-\pi_{\text{latent}}$ indicate ProFITi where the permutation of all the observations (including channels and time points) are set on the latent embedding. Specifically, we pass $h$ through an MLP and selected the permutation by sorting its output. Significant difference in results is not observed because the ordering in lower triangular matrix can be seen as a Bayesian network, and the graph with the triangular matrix as adjacency is a full directed graph, and all of them induce the same factorization. Also, a triangular linear map $z \mapsto \mathrm{L}z$ to a distribution can describe any covariance matrix $\Sigma$ via a Cholesky decomposition $\Sigma = \mathrm{L}^T\mathrm{L}$, as $\rho(\mathrm{L}z) = ce^{-\frac{1}{2}z^T\mathrm{L}^T\mathrm{L}z}$.

### H.3 VARYING OBSERVATION AND FORECAST HORIZONS

In Table 7, we compare ProFITi with two next best models, GraFITi+ and Neural Flows. Our evaluation involves varying the observation and forecast horizons on the Physionet'12 dataset. Furthermore, we also compare with ProFITi-$A^{\text{tri}}$+$A^{\text{reg}}$, wherein the triangular attention mechanism in ProFITi is replaced with a regularized attention mechanism.

ProFITi exhibits superior performance compared to both Neural Flows and GraFITi+, demonstrating a significant advantage. We notice that when we substitute $A^{\text{tri}}$ with $A^{\text{reg}}$; this change leads to a degradation in performance as the forecast sequence length increases. Also, note that the run time for computing $A^{\text{reg}}$ and its determinant is an order of magnitude larger than that of $A^{\text{tri}}$. This is because, it requires $\mathcal{O}(K^3)$ complexity to compute spectral radius $\sigma(A)$ and determinant of $A^{\text{reg}}$, whereas computing determinant of $A^{\text{tri}}$ requires $\mathcal{O}(K)$ complexity.

Additionally, we see that as the sequence length increases, there is a corresponding increase in the variance of the NJNL. This phenomenon can be attributed to the escalating number of target values ($K$), which increases with longer sequences. Predicting the joint distribution over a larger set of target values can introduce noise into the results, thereby amplifying the variance in the outcomes. Whereas for the GraFIT+ and Neural Flows it is not the case as they predict only marginal distributions. Further, as expected the NJNL of all the models decrease with increase in sequence lengths as it is difficult to learn longer horizons compared to short horizons of the forecast.

In Figure 9, we show the qualitative performance of ProFITi. We compare the trajectories predicted by ProFITi by random sampling of $z$ with the distribution predicted by the GraFITi+ (next best model).

### H.4 EXPERIMENT WITH VARYING $\epsilon$ IN EQ. 6

Here, we show the performance of ProFITi with varying $\epsilon$ in Eq. 6. We varied the $\epsilon$ among $\{0.001, 0.01, 0.1, 1, 10\}$. Results are presented in Table 8. Other than $\epsilon$, best hyperparameters used to obtain results for Physionet'12 in Table 3 are used. We observe that for smaller values of $\epsilon$ results does not change significantly. But with larger values, ProFITi performs very poorly. In all our experiments we set $\epsilon = 0.1$.

Table 8: Varying $\epsilon$ in Eq. 6. Used Physionet'12 dataset, evaluation metric NJNL, lower the best.

| $\epsilon$ | Physionet'12 |
|---|---|
| 0.001 | -0.753±0.046 |
| 0.01 | -0.698±0.033 |
| 0.1 | -0.766±0.038 |
| 1 | -0.443±0.096 |
| 10 | > 1e7 |

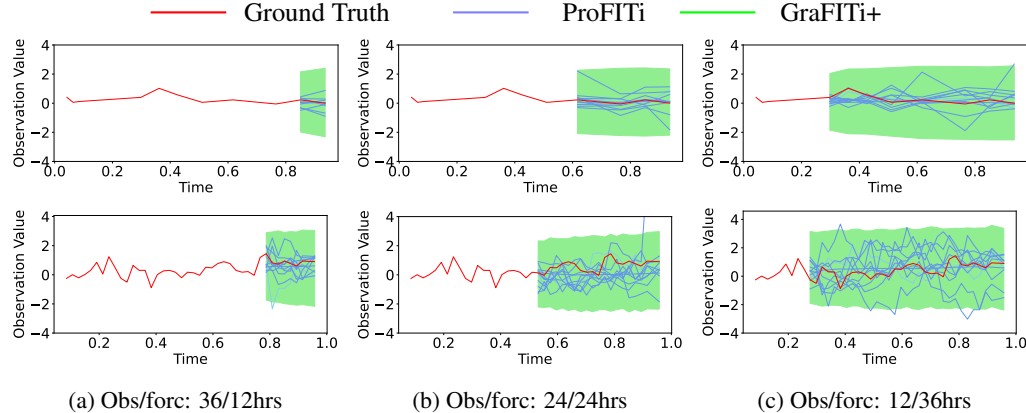

Figure 9: Demonstrating (10) trajectories generated using ProFITi for Physionet'12 dataset.

Table 9: Varying #observations in the time series. Physionet'12 dataset, evaluation metric NJNL.

| | % missing observations | | |
| --- | --- | --- | --- |
| | 10% | 50% | 90% |
| Neural Flow | 0.497±0.042 | 0.542±0.031 | 0.677±0.018 |
| GraFITi+ | 0.402±0.016 | 0.481±0.018 | 0.666±0.012 |
| ProFITi | **-0.141±0.036** | **0.077±0.012** | **0.336±0.033** |

### H.5 EXPERIMENT WITH VARYING NUMBER OF MISSING VALUES

Here, we experimented on Physionet'12 dataset with varying sparsity levels. We randomly removed $x\%, x \in \{10, 50, 900\}$ of observations in the series. Compared GraFITi+, Neural Flow and ProFITi. We observe that even with 90% missing values, ProFITi perform significantly better.

### H.6 EXPERIMENT WITH VARYING TIME SPARSITY

Table 10: Varying #observation events i.e., time points in the time series. Physionet'12 dataset, evaluation metric NJNL.

| | % missing observation events | | |
| --- | --- | --- | --- |
| | 10% | 50% | 90% |
| Neural Flow | 0.528±0.037 | 0.578±0.048 | 0.858±0.006 |
| GraFITi+ | 0.469±0.032 | 0.520±0.022 | 0.767±0.004 |
| ProFITi | **-0.106±0.112** | **-0.160±0.056** | **0.128±0.056** |

We use Physionet'12 dataset to experiment on varying number of observation events i.e. time points. We randomly removed $x\%, x \in \{10, 50, 900\}$ of observation events in the series and compared GraFITi+, Neural Flow and ProFITi. Again, we observe that even with 90% of time points missing, ProFITi perform significantly better.

## I HYPERPARAMETERS SEARCHED

Following the original works of the baseline models, we search the following hyperparameters:

**HETVAE (Shukla & Marlin, 2022)** :

- Latent Dimension: {8, 16, 32, 64, 128}
- Width : {128,256,512}
- # Reference Points: {4, 8, 16, 32}

- # Encoder Heads: $\{1, 2, 4\}$
- MSE Weight: $\{1, 5, 10\}$
- Time Embed. Size: $\{16, 32, 64, 128\}$
- Reconstruction Hidden Size: $\{16, 32, 64, 128\}$

**GRU-ODE-Bayes (De Brouwer et al., 2019)** :

- solver: $\{$euler, dopri5$\}$
- # Hidden Layers: $\{3\}$
- Hidden Dim.: $\{64\}$

**Neural Flows (Biloš et al., 2021)** :

- Flow Layers: $\{1, 4\}$
- # Hidden Layers: $\{2\}$
- Hidden Dim.: $\{64\}$

**CRU (Schirmer et al., 2022)** :

- # Basis: $\{10, 20\}$
- Bandwidth: $\{3, 10\}$
- lsd: $\{10, 20, 30\}$

**CNF+** :

- # Attention layers: $\{1,2,3,4\}$
- # Projection matrix dimension for attention: $\{32,64,128,256\}$

**GraFITi+ (Yalavarthi et al., 2023)** :

- # layers: $\{2, 3, 4\}$
- # MAB heads: $\{1, 2, 4\}$
- Latent Dim.: $\{32, 64, 128\}$

**ProFITi (Ours)** :

- # Flow layers: $\{8, 9, 10\}$
- $\epsilon$: $\{0.1\}$
- Latent Dim.: $\{32, 64, 128, 256\}$

