has been only two works that apply invertible attention in Normalizing Flows. Sukthanker et al. (2022) proposed invertible attention mechanism by adding the identity matrix to softmax attention. However, the major disadvantage of this mechanism is that softmax yields only positive values in the attention matrix and does not allow learning negative covariances. Zha et al. (2021) introduced residual attention similar to invertible residual flow (Chen et al., 2019; Behrmann et al., 2019). It has similar problems related to residual flows (Behrmann et al., 2019) such as lack of analytical inverse making it extremely slow to infer due to requirement of numerical methods. Additionally, these attention mechanisms often have problems with computing the determinant of the attention matrix as it has a complexity of $\mathcal{O}(K^3)$ with $K$ being the sequence length. We provide the summary of the related work in Table 1.

## 3 PROBLEM SETTING & ANALYSIS

**The IMTS Forecasting Problem.**    An **irregularly sampled multivariate times series with missing values** (called briefly just **IMTS** in the following), is a finite sequence $x^{\text{obs}} = \left( (t_i^{\text{obs}}, c_i^{\text{obs}}, o_i^{\text{obs}}) \right)_{i=1:I}$ of unique triples, where $t_i^{\text{obs}} \in \mathbb{R}$ denotes the time, $c_i^{\text{obs}} \in \{1, ..., C\}$ the channel and $o_i^{\text{obs}} \in \mathbb{R}$ the value of an observation, $I \in \mathbb{N}$ the total number of observations across all channels and $C \in \mathbb{N}$ the number of channels. Let $\text{TS}(C) := (\mathbb{R} \times \{1, \dots, C\} \times \mathbb{R})^*$ denote the space of all IMTS with $C$ channels.[1]

An **IMTS query** is a finite sequence $x^{\text{qu}} = \left( (t_k^{\text{qu}}, c_k^{\text{qu}}) \right)_{k=1:K} \in \text{Q}(C) := (\mathbb{R} \times \{1, \dots, C\})^*$ of just time points and channels (also unique), a sequence $y \in \mathbb{R}^K$ we call an answer and denote by $\text{QA}(C) := \{ (x^{\text{qu}}, y) \in \text{Q}(C) \times \mathbb{R}^* \mid |x^{\text{qu}}| = |y| \}$ the space of all queries and possible answers.

The **IMTS probabilistic forecasting problem** then is, given a dataset $\mathcal{D}^{\text{train}} := \left( (x^{\text{obs}}, x^{\text{qu}}, y) \right)^* \in \text{TS}(C) \times \text{

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

$$\mathrm{EL}(y_k; x_k) := y_k \cdot \mathrm{NN}^{\mathrm{sca}}(x_k) + \mathrm{NN}^{\mathrm{trs}}(x_k) \quad (10)$$

where $\mathrm{NN}^{\mathrm{sca}}$ and $\mathrm{NN}^{\mathrm{trs}}$ are neural networks for scaling and translation. $\mathrm{NN}^{\mathrm{sca}}$ is equipped with a $\exp(\tanh)$ output function to make it positive and bounded, guaranteeing inverse. We combine all three layers from eq. 6, 8, and 10 to a block

$$\text{profiti-block}(y; x) := \mathrm{Shiesh}(\mathrm{EL}(\mathrm{ISA}(z; x); x)) \quad (11)$$

We add a transformation layer with slope fixed to 1 as initial encoding on the $y$-side of the model. See fig. 3 for an overview of its architecture.

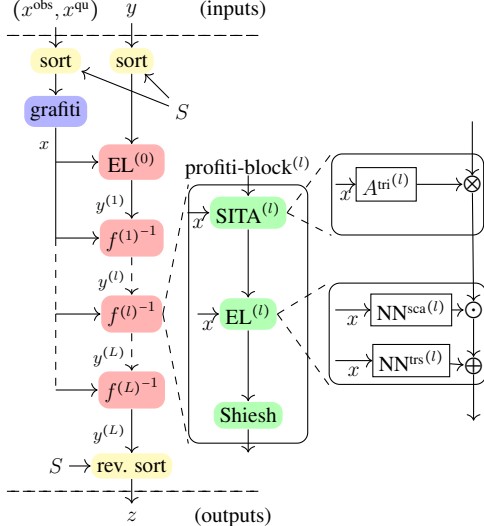

Figure 3: ProFITi architecture. $\otimes$: dot product, $\odot$: Hadamard product, and $\oplus$: addition.

**Query embedding.** As discussed in Section 4, for probabilistic time series forecasting we have to condition on both, the past observations $x^{\mathrm{obs}}$ and the time point/channel pairs $x^{\mathrm{qu}}$ of interest that are queried. While in principle any equivariant encoder could be used, an encoder that leverages the relationships between those two pieces of the conditioner is crucial. We use GraFITi (Yalavarthi et al., 2023), a graph based deterministic equivariant forecasting model for IMTS that provide state-of-the-art performance as encoder

$$(x_1, \ldots, x_K) := \mathrm{GraFITi}(x_1^{\mathrm{qu}}, \ldots, x_K^{\mathrm{qu}}, x^{\mathrm{obs}})$$

The encoded conditioners $x_1, \ldots, x_K$ then are fed into ProFITi as in eq. 11. The Grafiti encoder is trained end-to-end within the Profiti model, we did not pretrain it.

**Training.** We train the ProFITi model $\hat{p}$ for the normalized joint negative log-likelihood loss (NJNL; eq. 1) which is written in terms of the transformation $f^{-1}(\cdot; \cdot; \theta)$ of the normalizing flow and its parameters $\theta$ yields:

$$\ell^{\mathrm{NJNL}}(\theta) := \ell^{\mathrm{NJNL}}(\hat{p}; p) \quad (12)$$

$$= - \mathop{\mathbb{E}}_{(x^{\mathrm{obs}}, x^{\mathrm{qu}}, y) \sim p} \frac{1}{|y|} \left( \sum_{k=1}^{|y|} p_Z(f^{-1}(y; x^{\mathrm{obs}}, x^{\mathrm{qu}}; \theta)_k) - \log \left| \det \left( \frac{\partial f^{-1}(y; x^{\mathrm{obs}}, x^{\mathrm{qu}}; \theta)}{\partial y} \right) \right| \right)$$

## 7 EXPERIMENTS

### 7.1 EXPERIMENT FOR NORMALIZED JOINT NEGATIVE LOG-LIKELIHOOD

**Datasets** We use 3 publicly available real-world medical IMTS datasets: MIMICIII (Johnson et al., 2016), MIMICIV (Johnson et al., 2021), and Physionet (Silva et al., 2012), for evaluating the ProFITi architecture. Datasets contain ICU patient records collected over 48 hours. The pre-processing procedures outlined in (Yalavarthi et al., 2023; Scholz et al., 2023; Biloš et al., 2021; De Brouwer et al., 2019) were applied to MIMIC-III, and MIMIC-IV datasets. Consequently, observations in MIMIC-III and MIMIC-IV were rounded to intervals of 30 minutes and 1 min, respectively. As for Physionet'12, the dataset was processed according to (Yalavarthi et al., 2023; Che et al., 2018; Cao et al., 2018; Tashiro et al., 2021) and obtain hourly observations.

**Baseline Models** ProFITi is compared with 3 probabilistic IMTS forecasting models: CRU (Schirmer et al., 2022), Neural-Flows Biloš et al. (2021), GRU-ODE-Bayes De Brouwer et al.

Table 3: Normalized Joint Negative Log-likelihood (NJNL), lower the best. Best is presented in bold. OOM indicates out of memory error. Likelihood ↑ shows percentage gain and actual gain in Normalized Joint Likelihood $e^{-\text{NJNL}}$ (not log-level NJNL) compared to the next best model.

|  | Physioinet'12 | MIMIC-III | MIMIC-IV |
|---|---|---|---|
| GPR | 1.367±0.074 | 3.146±0.359 | 2.789±0.057 |
| HETAVE | 0.561±0.012 | 0.794±0.032 | OOM |
| GRU-ODE | 0.501±0.001 | 0.837±0.012 | 0.823±0.318 |
| Neural-Flows | 0.496±0.001 | 0.835±0.014 | 0.689±0.087 |
| CRU | 0.741±0.001 | 1.090±0.001 | OOM |
| CNF | 1.057±0.007 | 1.123±0.005 | 1.041±0.010 |
| GraFITi+ | 0.367±0.021 | 0.695±0.019 | 0.287±0.040 |
| ProFITi (ours) | **-0.766±0.038** | **-0.240±0.068** | **-1.856±0.051** |
| Likelihood ↑ | 210% (3.1×) | 150% (2.5×) | 755% (8.5×) |

(2019). We also extend the state-of-the-art deterministic forecasting model GraFITi (Yalavarthi et al., 2023) for the probabilistic setup by learning variance of the normal distribution along with the mean and call it as GraFITi+. Because one can often use interpolation models for the forecasting task, we include HETVAE, state-of-the-art probabilistic interpolation model, for the comparison. Further, we apply Multi-task Gaussian Process Regression model (GPR) (Dürichen et al., 2015). Since, we could not find any CNF model that works for dynamic size and conditioning input, we adapt continuous normalizing flow model (CNF) from (Biloš & Günnemann, 2021) to make it conditional as explained in Section 4. We set L2 regularized multivariate attention as $g$ in eq. 4.

**Protocol** We split the dataset into Train, Validation and Test in 70:10:20 ratio respectively. We select the hyperparameters using validation dataset on 10 random hyperparameter sets. We run for 5 iterations with random seeds on the test dataset with the chosen hyperparameters. Following (Biloš et al., 2021; De Brouwer et al., 2019; Yalavarthi et al., 2023), we set observation range as the first 36 hours and forecast the next 3 time steps. All the models were experimented using the PyTorch library on GeForce RTX-3090 and 1080i GPUs. We compare the models for Normalized Joint Negative Log-likelihood (NJNL) loss (eq. 1). Except for GPR, CNF and ProFITi, we take the average of the marginal negative log-likelihoods of all the observations in a series to compute NJNL for that series.

**Results** Table 3 demonstrates the Mean Normalized Joint Negative Log-likelihood for all the datasets (lower the best). Best results are presented in bold. ProFITi outperforms all the prior approaches with significant margin on all the three datasets. The next best performing model in Physionet'12 and MIMIC-III dataset is GraFITi+ whereas for MIMIC-IV it is NeuralFlows. We note that although GPR considers joint likelihoods, it performs poorly because of having very few parameters. We cannot run CRU model for MIMIC-IV in our GPU with 48GB memory as it is giving memory errors. Surprisingly, despite being a flow model, CNF did not perform as well in our tasks. *The performance gains shown in the table are not on NJNL but normalized joint likelihoods ($e^{-\text{NJNL}}$), because of 0 in the space of NJNL.*

### 7.2 AUXILIARY EXPERIMENT FOR MARGINALS AND POINT FORECAST

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

**GRU-ODE-Bayes (De Brouwer et al., 2019)** :

- solver: $\{\text{euler, dopri5}\}$
- # Hidden Layers: $\{3\}$
- Hidden Dim.: $\{64\}$

**Neural Flows (Biloš et al., 2021)** :

- Flow Layers: $\{1, 4\}$

- # Hidden Layers: {2}
- Hidden Dim.: {64}

**CRU (Schirmer et al., 2022)** :

- # Basis: {10, 20}
- Bandwidth: {3, 10}
- lsd: {10, 20, 30}

**CNF+** :

- # Attention layers: {1,2,3,4}
- # Projection matrix dimension for attention: {32,64,128,256}

**GraFITi+ (Yalavarthi et al., 2023)** :

- # layers: {2, 3, 4}
- # MAB heads: {1, 2, 4}
- Latent Dim.: {32, 64, 128}

**ProFITi (Ours)** :

- # Flow layers: {8, 9, 10}
- $\epsilon$: {0.1}
- Latent Dim.: {32, 64, 128, 256}