# OpenReview forum: "ProFITi: Probabilistic Forecasting of Irregular Time Series via Conditional Flows"
_ICLR.cc/2024/Conference — Submitted to ICLR 2024_

### Official Review · Reviewer_jB4x · 2023-10-25

**Soundness:** 3 good
**Presentation:** 3 good
**Contribution:** 3 good
**Rating:** 6
**Confidence:** 4

**Summary:**

Authors posit a new normalizing flow model for time series forecasting. They introduce two key components, the Shiesh activation function and SITA.

Edit: After reading author responses and reviewer comments, I have decided to maintain my score.

**Strengths:**

Shiesh seems to have good performance for normalizing flow models. I wonder if the authors have done experimentation with regards to the activation function in particular for other normalizing flow models?

The ablation studies conducted are quite thorough with respect to each of the components, and authors offer solid error bars for each of the results.

Proposed Shiesh activation function and SITA are both well formulated and grounded. These should also be able to be extended to alternative flow models.

**Weaknesses:**

Figure 1: I'm not sure what this is trying to illustrate, since you're comparing a target bimodal distribution against two other linear gaussian models, and doesn't do much to highlight the novelty of ProFITi.

Experiments seem to only consist of ECG based dataset, which is heavily periodic and consists of similar patterns. Would be interesting to see other datasets here.

Minor typo: "Both, Leaky" doesn't need a comma.

**Questions:**

What activation function is used in ProFITi-Shiesh?
Authors also mention Leaky ReLU but do not note where it is.

---

> ### Author Response · Authors · 2023-11-18
> **Response to reviewer jB4x**
>
> We thank you for your positive review. We address your concerns below:
>
> ## w1. Figure 1: I'm not sure what this is trying to illustrate, since you're comparing a target bimodal distribution against two other linear gaussian models, and doesn't do much to highlight the novelty of ProFITi.
> - Existing models for probabilistic IMTS forecasting predict only Gaussian distributions (ex. HETVAE, CRU, GRU-ODE, Neural Flows). So fig. 1 is a simple illustration of the goals and achievements of our model ProFITi: to quantify uncertainty in a richer manner. We edited the introduciton to make this clearer.
>
> ## w2. Experiments seem to only consist of ECG based dataset, which is heavily periodic and consists of similar patterns. Would be interesting to see other datasets here.
> - We agree. We added experiments on USHCN, an IMTS climate dataset that also has been previously used in the literature (De Brouwer et al. 2019, Schirmer et al. 2022, Yalavarthi et al. 2023) to section 7.
>
> ## w3. Minor typo: "Both, Leaky" doesn't need a comma.
> - Fixed. Thanks!
>
> ## q1. What activation function is used in ProFITi-Shiesh? Authors also mention Leaky ReLU but do not note where it is.
> - ProFITi-Shiesh does not contain any activation function in the flow
>   model. It is used as ablation to demonstrate the use of a non-linear
>   activation function in ProFITi.
> - Leaky ReLU with very small slope (0.01) for the negative values suffers
>   from the vanishing gradient problem. Therefore no results
>   are shown. We added this information to Appendix H.1.

---

### Official Review · Reviewer_qpff · 2023-10-30

**Soundness:** 3 good
**Presentation:** 3 good
**Contribution:** 3 good
**Rating:** 6
**Confidence:** 3

**Summary:**

The work proposes a probabilistic forecasting for irregular time series using conditional normalizing flows.

**Strengths:**

The probabilistic forecasting of irregular timeseries is an important problem and the use of normalizing flow for this scope is interesting.

Introduction of a new activation function having the characteristics useful for the normalizing flows.

**Weaknesses:**

In my opinion could be interesting to assess the behavior of the proposed model in several conditions in terms of percentage of missing values, sparsity of the time steps. In fact the irregular time series are very important and to give researcher an evaluation of these aspects could help to understand the quality/limitations of the proposed work.

The authors use likelihood to assess the quality of the forecast also for marginal and point forecast. In that case I think could be useful to have other metrics as RMSE, MAPE or other that is more related to the prediction error.

The computational effort of proposed solution is not provided and is not compared to other possible solutions. Some results are provided in appendix but only for a particular dataset.

**Questions:**

§3 - IMTS Query section, in the definition of QA(C), what is the meaning of $|x^{qu}|=|y|$? After, when you talk about NJNL, looks like $|.|$ means lenght of, but authors should describe clearlry the symbols they use.

In the IMTS probabilistic forecasting problem
"(with $min_k t_{n,k}^{qu} > max_i t_{n,i}^{obs}$)" --> the reader is lead to think that $t_{n,k}^{qu}$ is the k-th observation of the n-th query and the same for $t_{n,i}^{obs}$ but a clarification would be better.

What is the meaning of $\pi^{-1}$ in equation 4?

In eq. 7 $X_{:,1:|X|-1}$ is the matrix X except the last column?
In eq. 7 $X_{:,|X|}$ is the last column of matrix X ?

In the protocol authors said that they use the first 36 hours of observations and forecast next 3 time steps. These time step is, looking at Appendix A, 1 hour for Physionet, 30minute for MIMIC-III, 1 minute for MIMIC-IV. Is this right?
Moreover, is there a way to indicate the time sparsity of considered timeseries in oder to understand how the timeseries considered are not uniformly spaced?

The section 6 and figure 3 should be improved in order to make clearer the architecture.
In equation 11 the $x_k$ is the $h$ of figure 3?
What is the meaning of S in fig. 3?

In Fig. 9 the Grafiti+ returns only a region an not the trajectories? Is it possible to compare the confidence regions of ProFITi and GraFITi+?

Some general comments (also present in the Weakness section)
In my opinion could be interesting to assess the behavior of the proposed model in several conditions in terms of percentage of missing values, sparsity of the time steps. In fact the irregular time series are very important and to give researcher an evaluation of these aspects could help to understand the quality/limitations of the proposed work.

The authors use likelihood to assess the quality of the forecast also for marginal and point forecast. In that case I think could be useful to have other metrics as RMSE, MAPE or other that is more related to the prediction error.

The computational effort of proposed solution is not provided and is not compared to other possible solutions. Some results are provided in appendix but only for a particular dataset.

---

> ### Author Response · Authors · 2023-11-18
> **Response to Reviewer qpff**
>
> Thank you for the detailed review. We address the concerns below:
>
> ## w1+q8. In my opinion could be interesting to assess the behavior of the proposed model in several conditions in terms of percentage of missing values, sparsity of the time steps. In fact the irregular time series are very important and to give researcher an evaluation of these aspects could help to understand the quality/limitations of the proposed work.
> - We agree. We added an ablation study with varying sparsity levels
>   (time sparsity and missing values) in Appendix H.5, H.6. Our findings
>   show that even in an extremely sparse scenario, where 90% of samples
>   (or timepoints) are missing, ProFITi consistently outperforms both
>   Neural Flows and GraFITi+.
>
> ## w2+q9. The authors use likelihood to assess the quality of the forecast also for marginal and point forecast. In that case I think could be useful to have other metrics as RMSE, MAPE or other that is more related to the prediction error.
> - We **do use MSE** to evaluate models w.r.t. point forecasts (tab. 4).
>
> ## w3+q10. The computational effort of proposed solution is not provided and is not compared to other possible solutions. Some results are provided in appendix but only for a particular dataset.
> - In the updated version, we included the **runtime per epoch
>    for all models and datasets** in Table 3.
> - Our main claim is to provide a **more accurate model**, not to provide a faster
>   model (contribution 5). Thus we gave more room to evaluation w.r.t. accuracy
>   but w.r.t. scalability.
>
> ## q1. §3 - IMTS Query section, in the definition of $\text{QA(C)}$, what is the meaning of |x^\text{qu}| = |y|?
> - yes, |.| denotes the length of a sequence. We added a brief section on notation (appendix A).
>
> ## q2. In the IMTS probabilistic forecasting problem (with $\min_k t^\text{qu}_{n,k} > \max_i t^\text{obs}_{n, i}$) --> the reader is lead to think that is the k-th observation of the n-th query
> - You are right. We added a respective note to section 3.
>
> ## q3. What is the meaning of $\pi^{-1}$ in equation 4?
> - for permutations $\pi$, one often writes $x^{\pi}$ to denote the application of the permutation $\pi$ to vector x.
> - $\pi^{-1}$ is permutations that are inverse w.r.t $\pi$. $(x^\pi)^{\pi^{-1}} = x$
> - we added a brief section on notation (appendix A).
>
> ## q4. In eq. 7 $X_{:, 1:|X|-1}$ is the matrix X except the last column? In eq. 7 $X_{:, |X|}$ is the last column of matrix X ?
> - yes. We added this, too, to the notation section (appendix A).
>
> ## q5. In the protocol authors said that they use the first 36 hours of observations and forecast next 3 time steps. These time step is, looking at Appendix A, 1 hour for Physionet, 30minute for MIMIC-III, 1 minute for MIMIC-IV. Is this right? Moreover, is there a way to indicate the time sparsity of considered timeseries in oder to understand how the timeseries considered are not uniformly spaced?
> - Yes, the observation intervals are data set specific.
> - We added an additional column "time sparsity" to table 5 that quantifies
>   the sparsity of the time series (as proxy measure for its irregularity).
>
> ## q6. The section 6 and figure 3 should be improved in order to make clearer the architecture. In equation 11 the $x_k$ is the h of figure 3? What is the meaning of S in fig. 3?
> - Yes, $h_k$ are the encoded $x_k$ and replace $x_k$ everywhere (as stated
>   below their definition in paragraph "Query embedding": "take the roles
>   of the $x_k$").
> - $S$ is the sorting criterion introduced at the end of section 5.
> - We added references for all symbols in fig. 3 to their respective definitions to make navigation between formulas and the diagram faster.
>
> ## q7. In Fig. 9 the Grafiti+ returns only a region an not the trajectories? Is it possible to compare the confidence regions of ProFITi and GraFITi+?
> - Since ProFITi provides non-parametric distribution, it is difficult to plot the confidence regions for multiple points simultaneuously. Hence, although it is interesting, we do not want to pursue the idea.

---

> > ### Comment · Reviewer_qpff · 2023-11-22
> >
> > After reading other reviews, authors replies, and revised manuscript, I decided to improve my initial score.

---

### Official Review · Reviewer_DHMB · 2023-10-31

**Soundness:** 3 good
**Presentation:** 1 poor
**Contribution:** 2 fair
**Rating:** 5
**Confidence:** 4

**Summary:**

The paper focuses on probabilistic forecasting of irregularly sampled multivariate time series with missing values. The authors propose to use conditional continuous normalizing flow to construct the distribution instead of making an assumption on the target distribution as done in the literature. Moreover, they provide a novel invertible equivariant transformation, and a novel non-linear, invertible, differentiable activation function, which can be used in normalizing flows. Finally, they conduct extensive experiments on three real-world IMTS datasets and show that the proposed model (PROFITI) outperforms baselines in terms of normalized joint negative log-likelihood.

**Strengths:**

The exploration of predicting changes in the joint distribution of time series is an important and valuable problem, which is crucial for downstream tasks in various domains. While previous literature has predominantly focused on regular time series, the paper's contribution in addressing the prediction of joint distributions in irregular time series is commendable. Moreover,  the paper showcases a significant amount of effort and extensive work. The authors present compelling evidence of the effectiveness of their proposed method. The results indicate that the approach performs well in predicting the joint distribution.

**Weaknesses:**

1. The requirement of permutation invariance, as proposed by the authors, is questionable in time series analysis and may not be suitable for this domain. The authors argue that a density model should produce equivalent density values when the outputs are swapped, which is a reasonable expectation in the context of static generative models. However, in the time series setting, where the joint distribution of variables $y_1,...,y_K$ occurring at different time steps is considered, the presence of serial dependencies becomes crucial. Unfortunately, the permutation invariant requirement, which treats the order of data points as interchangeable, risks disrupting these vital temporal dependencies. In contrast, if the objective is to forecast the joint distribution of variables occurring at the same time step, the permutation invariant requirement might be deemed necessary.

2. The rationale behind utilizing self-attention as the vector field in the proposed work is not clear. It seems redundant to introduce self-attention into the vector field, given that neural networks inherently possess permutation invariance properties with respect to the input. Therefore, a more explicit justification is needed to understand the motivation behind incorporating self-attention in this context. Moreover, it is worth noting that existing literature, such as [1], has already proposed the use of conditional normalizing flow models for forecasting the joint distribution of time series. Therefore, it is crucial to highlight the distinctions between the proposed work and the existing literature that necessitate the use of self-attention and the introduction of a more complex invertible self-attention mechanism.


[1] Rasul, K., Sheikh, A. S., Schuster, I., Bergmann, U., & Vollgraf, R. (2020). Multivariate probabilistic time series forecasting via conditioned normalizing flows. arXiv preprint arXiv:2002.06103.

**Questions:**

1. In the paragraph “Invariant conditional normalizing flows,” the authors claim that x is the predictor, which can be grouped into K elements and common elements. The statement confuses me. It is not clear why x contains common elements. Could the authors provide a concrete example of the general setting and explain the statement in the paragraph?

2. What is the meaning of the L.H.S. in Eqn. (4)? There is a superscript $\pi^{-1}$ on the L.H.S. in Eqn. (4), what does it mean?

3. The choice of $\epsilon$ in Eqn. (6) should affect the training results. Should it be determined before training? Or should it be termed as a hyperparameter during training? Should there be an investigation about the choice of $\epsilon$?

4. Why use the GraFITi model to encode the historical data? Compared to the prevalent model used to model irregular time series, such as GRU-ODE-Bayes [2] or  Neural CDE [3], what is the advantage of GraFITi?

5. Most literature use CRPS and CRPS_sum to evaluate the performance, e.g., [1, 4-6]. Why authors do not follow the literature?

[2] De Brouwer, E., Simm, J., Arany, A., & Moreau, Y. (2019). GRU-ODE-Bayes: Continuous modeling of sporadically-observed time series. Advances in neural information processing systems, 32.

[3] Kidger, P., Morrill, J., Foster, J., & Lyons, T. (2020). Neural controlled differential equations for irregular time series. Advances in Neural Information Processing Systems, 33, 6696-6707.

[4] Salinas, D., Bohlke-Schneider, M., Callot, L., Medico, R., & Gasthaus, J. (2019). High-dimensional multivariate forecasting with low-rank gaussian copula processes. Advances in neural information processing systems, 32.

[5] Salinas, D., Flunkert, V., Gasthaus, J., & Januschowski, T. (2020). DeepAR: Probabilistic forecasting with autoregressive recurrent networks. International Journal of Forecasting, 36(3), 1181-1191.

[6] Rasul, K., Seward, C., Schuster, I., & Vollgraf, R. (2021). Autoregressive denoising diffusion models for multivariate probabilistic time series forecasting. In International Conference on Machine Learning (pp. 8857-8868). PMLR.

---

> ### Author Response · Authors · 2023-11-17
> **Response to Reviewer DHMB**
>
> Thanks for your detailed review. We address the comments below:
>
> ## w1. The requirement of permutation invariance, as proposed by the authors, is questionable in time series analysis
>
> - Permutation invariance is crucial **w.r.t. channels**: it should not matter,
>   if you query for channel 1 or channel 2 first.
> - Permutation invariance **w.r.t. time points** is not a necessary property
>   of a good model due to the causal aspects you mention. But it is a
>   **very successful property** of most state-of-the-art models in vision and
>   time series. This motivated us to search for a novel invertible variant
>   that can be used in conjunction with normalizing flows.
> - All queries carry the **time point as an attribute** ($q_k$ in the paper),
>   like positional encodings in transformers, so that the model can
>   detect and represent temporal dependencies, of course.
> - We added a respective paragraph to our problem analysis section 3.
>
> ## w2a. The rationale behind utilizing self-attention as the vector field in the proposed work is not clear. It seems redundant to introduce self-attention into the vector field, given that neural networks inherently possess permutation invariance properties with respect to the input.
> - Standard fully connected layers in neural networks are not permutation invariant: for example, if the weight matrix is W = [[1,0],[0,1]] the identity, then the output Wx = x varies under permutations, e.g., W[1,0] = [1,0], but W[0,1]= [0,1].
> - Invariance partially is crucial, partially is highly promising as explained above (see w1).
>
> ## w2b. existing literature, such as [1], has already proposed the use of conditional normalizing flow models for forecasting the joint distribution of time series
> - MAF (Rasul et al. 2020, [1]) is a model for **regularly sampled, fully observed time series** data, while ours is for **irregularly sampled time series with missing values** (also see table 1).
> - Forecasting queries for fully observed, regularly sampled data **always have the same size**, thus any (conditional) normalizing flow model can be used.
> - But forecasting queries for irregularly sampled time series with missing data **have varying sizes** (both w.r.t. to missing time points and missing observation values), and thus standard normalizing flow models are not applicable (column "joint" in table 1). This is one of the fundamental research problems we aimed to solve in our paper. Combining attention (allowing flexible size) with invertibility (to be used in a normalizing flow) is how we solved the problem. In consequence, MAF cannot predict joint distributions over channels and time points for any of the datasets in our evaluation.
> - We added this delineation to the related work section 2.
>
> ## q1. In the paragraph “Invariant conditional normalizing flows,” the authors claim that $x$ is the predictor, which can be grouped into $K$ elements and common elements. The statement confuses me. It is not clear why $x$ contains common elements. Could the authors provide a concrete example of the general setting and explain the statement in the paragraph?
> - in IMTS forecasting you have both inputs,
>   1. a variable number of queries $x^\text{qu}_1,...,x^\text{qu}_K$ (each a time point and channel), for each of which you want an output (the predicted value at that time point in that channel) and
>   2. the past observed time series $x^\text{obs}$ (see section 3, "IMTS probabilistic forecasting problem").
>   So here: $(x_1,...x_K):= (x^\text{qu}_1,...,x^\text{qu}_K)$, $x^\text{com} := x^\text{obs}$.
> - we added this example early in section 4.
>
> ## q2. What is the meaning of the L.H.S. in Eqn. (4)? There is a superscript $\pi^{-1}$ on the L.H.S. in Eqn. (4), what does it mean?
> - for permutations $\pi$, one often writes $x^{\pi}$ to denote the application of the permutation $\pi$ to vector $x$.
> - $\pi^{-1}$ is also a permutation which is inverse of $\pi$.
> - here it means, that $f$ is equivariant: if you permute inputs $z$ and $x$, then the outputs of $f$ are permuted the same way.
> - we added a brief section on notation (Appendix A).
>
> ## q3. The choice of $\epsilon$ in Eqn. (6) should affect the training results. Should it be determined before training? Or should it be termed as a hyperparameter during training? Should there be an investigation about the choice of?
> - yes, you are right, it does. It is a hyperparameter.
> - we clarified this next to Eq. 6.
> - we use $A^\text{tri}$ (Eq. 9 of revised version) as a vector field in all our experiments and set $\epsilon = 0.1$. It is also shown in Figure 2. We corrected the typo in Eq. 15 in the revised version (from ISA to SITA) which could have caused the confusion.
> - we added an ablation study for varying $\epsilon$ in $A^{\text{tri}}$ (appendix H.4).

---

> > ### Author Response · Authors · 2023-11-17
> > **Response to Reviewer DHMB (continued)**
> >
> > ## q4. Why use the GraFITi model to encode the historical data? Compared to the prevalent model used to model irregular time series, such as GRU-ODE-Bayes [2] or Neural CDE [3], what is the advantage of GraFITi?
> > - GRU-ODE-Bayes, CDE and other models are forecasting models, not encoders. They yield for each query a scalar, the predicted value, not an embedding vector. While in principle it would be possible to use them as (scalar) encoders, due to their limitations to a single dimension we did not follow up on this idea.
> > - We opted for GraFITi, because
> >   1. it has been shown to outperform many other models, incl. GRU-ODE-Bayes, NeuralFlow, CRU etc. (see their table 2) and
> >   2. it is very fast.
> > - We added this justification to section 6 / subsection "query embedding".
> >
> >
> > ## q5. Most literature use CRPS and CRPS_sum to evaluate the performance, e.g., [1, 4-6]. Why authors do not follow the literature?
> > - Most literature on forecasting **irregularly sampled time series**
> >   uses loglikelihood for comparison (De Brouwer et. al., 2019; Bilos et. al., 2021),
> >   we do the same to be directly comparable.
> > - You are right, that the literature on **predicting marginal distributions for
> >   regularly sampled time series** uses CRPS and/or CRPS_sum. But both measures
> >   do not cover joint distributions. So we did not use them.
> > - We add a delineation to the evaluation section 7.

---

> > ### Comment · Reviewer_DHMB · 2023-11-21
> >
> > Thanks for the authors' reply. For the response of w1, I agree that the permutation invariance is crucial w.r.t. channels. However, I am curious if it is too strong to use permutation invariance w.r.t. time points in time series forecasting, and I do not understand why permutation invariance w.r.t. time points is a very successful property of most state-of-the-art models in vision and time series. For example, if we have an irregular multivariate time series of two variables and want to estimate the joint distribution of $y_1^1, y_1^2$, and $y_2^1, y_3^2$ (the superscript indicates channel, and the subscript indicates time). The permutation w.r.t. channel implies that it can be decomposed as $$p(y_1^1, y_1^2)=p(y_1^1) p(y_1^2| y_1^1)= p(y_1^2) p(y_1^1| y_1^2). $$ However, the permutation w.r.t. time demonstrates that it can be decomposed as $$p(y_2^1, y_3^2)=p(y_2^1) p(y_3^2| y_2^1)= p(y_3^2) p(y_2^1| y_3^2).$$ What does $p(y_2^1| y_3^2)$ mean? Does it mean predicting the value at $t_2$ by conditioning the value at $t_3$? In addition, Rasul et al. (2020) use conditional normalizing flow (MAF) to estimate the joint distribution of time series at the next time point. It is a regular time series, but they never consider permutation invariance across various channels. Indeed, I hope authors can demonstrate the essential of permutation invariance w.r.t time through an experiment or theoretical analysis.

---

> > > ### Author Response · Authors · 2023-11-22
> > > **Permutation invariance of ProFITi**
> > >
> > > Thank you for your reply!
> > >
> > > Permutation invariance of the queries is not the same as the
> > > conditional decomposition. Permutation variant/dependent
> > > models have an inductive bias to learn correlations between
> > > the 1st, 2nd, 3rd element of a query. Permutation invariant
> > > models do not have this bias, they have to learn the correlation
> > > between queries based on the time point / the positional
> > > encoding alone.
> > >
> > > For decades researches have planted a strong inductive bias
> > > based on neighborhoods in their models (e.g., with convolutions),
> > > and we agree that this sounds plausible. However, since the
> > > application of transformers for vision (ViT, Dosovitskiy et al. 2021)
> > > attention mechanisms also have been used to learn those
> > > relations between positions, and these models have been highly
> > > successful, both providing state-of-the-art results and being
> > > highly cited by other researchers (ViT is cited 24823 times in
> > > 3 years).
> > >
> > > To the best of our knowledge, there is no way to prove that one approach is better than the other. But we provide extensive empirical evidence that enforcing permutation invariance also for joint probabilistic forecasting of irregular time series with missing values provides the best currently known predictions. We argue by analogy that this is not implausible because large parts of the ML community use similar models in similar contexts (e.g., vision transformers).
> > >
> > > References:
> > > - Dosovitskiy, Alexey; Beyer, Lucas; Kolesnikov, Alexander; Weissenborn,
> > >   Dirk; Zhai, Xiaohua; Unterthiner, Thomas; Dehghani, Mostafa;
> > >   Minderer, Matthias; Heigold, Georg; Gelly, Sylvain; Uszkoreit, Jakob:
> > >   "An Image is Worth 16x16 Words: Transformers for Image Recognition at Scale".
> > >   ICLR 2021.

---

### Official Review · Reviewer_JdGh · 2023-10-31

**Soundness:** 3 good
**Presentation:** 2 fair
**Contribution:** 3 good
**Rating:** 6
**Confidence:** 3

**Summary:**

The paper studies the problem of probabilistic forecasting of time series with irregular samples and missing values. Authors propose a new architecture based on conditional normalizing flows that can handle both, irregularity and missing values, and learns a joint distribution of the forecast targets. The proposed approach relies on the new invertible self-attention layer and a new activation function. The authors benchmark their method on 3 real-world datasets and showcase superior performance in terms of log-likelihood.

**Strengths:**

## Originality
To my best knowledge, the invertible self-attention layer and the activation functions are new
## Quality
* The presented approach can be learnt end-to-end without additional steps like solving ODE
* Experiments use a wide set of baselines
* Ablation study is performed
## Clarity
Authors do a good job motivating and explaining their design choices, but overall exposition is still pretty convoluted
## Significance
Experimental results show a significant improvement in log likelihood. New attention layer can be potentially used in other algorithms with normalizing flows, so the work can potentially have wide impact.

**Weaknesses:**

* My main concern is that it seems like most improvement comes from using GraFITi embeddings, which are not used by the baselines.
* Evaluation covers only three datasets from very similar domain
* The modelling limitations of the method are not clear. What kind of distributions can be modelled with the proposed flows?

**Questions:**

* Please adjust the abstract in line with ICLR formatting instructions
* It would be beneficial to study what kind of distributions can be modelled with the proposed flows.
* I wonder whether other methods would fare better if computed on the same GraFITi embeddings. Can you add a combination of other baselines with GraFITi and another option in the ablation study that doesn’t use it?

---

> ### Author Response · Authors · 2023-11-17
> **Response to Reviewer jdGH**
>
> We thank the reviewer for the valuable feedback. Responses to the weakness and questions are as follows
>
> ## w1. My main concern is that it seems like most improvement comes from using GraFITi embeddings, which are not used by the baselines.
> - Many baseline models such as NeuralFlows and GRU-ODE do not use
>   query encoders and thus cannot (easily) be extended to use
>   GraFITi.
> - But to disentangle lifts originating from GraFITi from those originating
>   from ProFITi, we added the model GraFITi+: the full GraFITi encoder
>   plus a simple uncertainty quantification model, a Gaussian (see table 4).
> - We added a note to section 7 Experiments.
>
> ## w2+q3.  Evaluation covers only three datasets from very similar domain
> - We agree. We added experiments on USHCN, an IMTS climate dataset
>   that also has been previously used in the literature (De Brouwer et al. 2019,
>   Schirmer et al. 2022, Yalavarthi et al. 2023) to section 7.
>
> ## w3+q2. The modeling limitations of the method are not clear. What kind of distributions can be modeled with the proposed flows?
> - ProFITi is a normalizing flow model, as such it is not limited to a distribution
>   with a fixed shape like the Gaussians.
> - However, quantifying the exact expressive power of normalizing flow
>   models is an active research topic of its own and beyond the scope of our
>   paper (see, e.g., Kong/Chaudhuri 2020):
> - We added a note to section 2 Literature review.
>
> ## q1. Please adjust the abstract in line with ICLR formatting instructions
> - Fixed. Thanks!
>
> Kong, Zhifeng, and Kamalika Chaudhuri. “The Expressive Power of a
>    Class of Normalizing Flow Models.” In Proceedings of the Twenty
>    Third International Conference on Artificial Intelligence and
>    Statistics, 3599–3609. PMLR, 2020. https://proceedings.mlr.press/v108/kong20a.html.

---

> > ### Comment · Reviewer_JdGh · 2023-11-22
> >
> > After reading other reviews and authors replies, I am inclined to maintain my score.

---

### Author Response · Authors · 2023-11-22

Dear reviewers,

thanks all of you for your many questions and points!
We tried hard to answer all of them and ensure that these
answers now can be found in the revised version of our paper.

Allow us to emphasize our main contribution, so it does not
get lost in the process: our paper is the first paper ever
to answer the following principled, important research
question: how can we forecast the joint distribution over
channels and time points if we have only sparse observations
(irregularly sampled and missing values). We develop
a model that clearly draws inspiration from existing
work, but in our opinion is not just a small, obvious
modification of existing work. We provide a benchmark
suite consisting of four data sets, our results, and our
code, so that future papers can build on top of ours
and improve it further. We are fully aware that our
model certainly is not the last model, that will solve this
task forever. But it is the first one and opens a new
and important research direction.

---

### Meta-Review · Area_Chair_rQX6 · 2023-12-06

**Metareview:**

The paper introduces a new flow-based architecture for irregularly sampled time series with missing data. Main contributions include a "sorted invertible triangular attention layer (SITA) parametrized by conditioning input, and an invertible activation function" and "permutation invariant model and designed to learn conditional permutation invariant structured distributions".

The reviewers raised several points about the rational of the approach (and its components) as well whether they are actually suitable for the task above.

**Justification For Why Not Higher Score:**

The reviewers were not entirely convinced about the paper, and rated it borderline.
I agree with the reviewers that the particular architecture choices (and their features, e.g. permuation invariance) remain quite unclear. I found it hard to grasp what the actual contribution of the paper is supposed to be. This is usually to little for ICLR.

**Justification For Why Not Lower Score:**

NA

---

### Decision · Program_Chairs · 2024-01-16

Reject